# A mechanical-assisted post-bioprinting strategy for challenging bone defects repair

Jirong Yang[1,2], Zhigang Chen[1,2], Chongjian Gao [1], Juan Liu[1], Kaizheng Liu [1], Xiao Wang[1,3], Xiaoling Pan[1,3], Guocheng Wang[1,2], Hongxun Sang[3], Haobo Pan [1,2,4], Wenguang Liu [5] & Changshun Ruan [1,2,4] ✉

Bioprinting that can synchronously deposit cells and biomaterials has lent fresh impetus to the field of tissue regeneration. However, the unavoidable occurrence of cell damage during fabrication process and intrinsically poor mechanical stability of bioprinted cell-laden scaffolds severely restrict their utilization. As such, on basis of heart-inspired hollow hydrogel-based scaffolds (HHSs), a mechanical-assisted post-bioprinting strategy is proposed to load cells into HHSs in a rapid, uniform, precise and friendly manner. HHSs show mechanical responsiveness to load cells within 4 s, a 13-fold increase in cell number, and partitioned loading of two types of cells compared with those under static conditions. As a proof of concept, HHSs with the loading cells show an enhanced regenerative capability in repair of the critical-sized segmental and osteoporotic bone defects in vivo. We expect that this post-bioprinting strategy can provide a universal, efficient, and promising way to promote cell-based regenerative therapy.

Reconstruction of challenging bone defects that cannot spontaneously heal, e.g., large-sized segmental and osteoporotic bone defects[1,2], remains a thorny problem in clinic. The difficulty is mainly attributed to the limited migration or weak regenerative potential of resident cells[3,4]. Notably, multiple cell types such as mesenchymal stem cells (MSCs) and endothelial cells (ECs), are involved in specific regenerative biofunctions e.g., osteogenesis and angiogenesis, playing vital roles in bone regeneration[5–8]. Regarding large-sized segmental bone defects, severely damaged surrounding tissues cause a lack of key cells to migrate and reside in the defect site, leading to no bridging and nonunion[9]. Osteoporosis is a typical metabolic bone disease, in which cells are senescent with weak regenerative potential. Once osteoporotic bone defects occur, it is hard for the senescent cells to support self-healing of bone tissues[10,11]. Whereas direct injection of related cells into the defect site results in suboptimal therapeutic outcomes[12], tissue-engineered bone substitutes consisted of biomaterial-based scaffolds and healthy cells bring a promising avenue for repair of challenging bone defects[1,13].

Despite tremendous progress in bone tissue engineering in the past decades, there are still limitations to be concerned. For instance, cells are conventionally seeded on the surface of scaffolds and then cultured in vitro until they migrate entirely inside scaffolds. Obviously, it is extremely time-consuming, and difficult to control cell distribution[14,15]. The emergence of three-dimensional (3D) bioprinting technology that can directly customize cell-laden constructs via mixing cells and biomaterials as bioinks provides new possibilities[16,17]. Due to its cost-effectiveness and ease of operation, extrusion bioprinting has become the most popular modality for fabrication of cell-laden constructs, followed by inkjet and light-based bioprinting[18,19]. Recently, several reports from our team and others have demonstrated extrusion-bioprinted cell-laden constructs with an enhanced effect on repair of bone defects[15,20–23]. However, in order to ensure fabrication of cell-laden constructs for bone repair, bioinks (essentially hydrogels[24])

[1]Research Center for Human Tissue and Organ Degeneration, Institute of Biomedicine and Biotechnology, Shenzhen Institute of Advanced Technology, Chinese Academy of Sciences, Shenzhen 518055, China. [2]University of Chinese Academy of Sciences, Beijing 100049, China. [3]Shenzhen Hospital, Southern Medical University, Shenzhen 518000, China. [4]The Key Laboratory of Biomedical Imaging Science and System, Chinese Academy of Sciences, Shenzhen 518055, China. [5]School of Materials Science and Engineering, Tianjin Key Laboratory of Composite and Functional Materials, Tianjin University, Tianjin 300350, China. ✉e-mail: cs.ruan@siat.ac.cn

must fulfill a number of "reasonable" requirements besides biocompatibility and bioactivity, which severely restrict the utilization of extrusion bioprinting for repair of challenging bone defects. For instance, a relatively reasonable viscosity of hydrogel precursor is required for extrusion: a high viscosity supports maintenance of homogeneous cell suspension in the printhead before printing; conversely, a low viscosity minimizes shear stress-driven cell damage from the nozzle during extrusion. Meanwhile, a relatively reasonable mechanical strength of hydrogels is also demanded: a high mechanical strength endows constructs with excellent self-supporting properties, beneficial for shape fidelity and large-sized fabrication; conversely, a low mechanical strength supports proliferation and migration of the embedded cells[25–27].

Herein, we propose a post-bioprinting strategy that cells are immediately loaded after scaffolds fabrication through a mechanical assistance (Fig. 1a). A type of heart-inspired hollow hydrogel-based scaffold (HHS) that can reversibly respond to external mechanical stimulations are used for active cell loading with minimal damage, similar to that the cardiac impulse attributes to systole and diastole. On basis of a customized combination of gelatin methacryloyl, Laponite nanoclay, and N-acryloyl glycinamide[28], HHSs were first fabricated by one-step coaxial printing without any supporting materials. Taken advantage of this hybrid ink, high-fidelity, and large-sized HHSs with uniform and intact hollow filament structures could be readily achieved and the structure was highly tunable. Interestingly, owing to the hollow structure, HHSs displayed attractive resilience, rapid shape

recovery, and exceptional fatigue resistance, which endowed them with an excellent mechanical responsiveness to load cells in a rapid, uniform, and precise manner. In addition, we verified the cell-laden HHSs for repair of critical-sized segmental and osteoporotic bone defects in rats, and the results demonstrated that the cell-laden HHSs exhibited a satisfying ability to heal the challenging bone defects. In all, this work offers a robust method for functional assembly of cells and biomaterials for tissue engineering.

## Results

### Fabrication of large-sized and sophisticated HHSs

HHSs were fabricated directly via extrusion printing using a coaxial needle without any supporting materials (Fig. 1a), which is convenient for building large-sized constructs with complicated shapes for tissue engineering (Fig. 1b). See Supplementary Movie 1 for the actual process of printing bone-shaped construct. After solidification by UV irradiation, stable HHSs were obtained with high shape fidelity and could suspend in water. The empty channels were clearly observed (Fig. 1c), indicating the feasibility of hollow structures in HHSs. To fabricate HHSs, hydrogel inks with excellent printability, appropriate mechanical behaviors as well as biocompatibility were indispensable. According to our previous report[28], the hybrid inks of gelatin methacryloyl/laponite nanoclay/N-acryloyl glycinamide (Supplementary Fig. 2) demonstrated excellent printability and sufficient mechanical properties for coaxial printing of microtubes for regeneration of tubular tissue. In this study, to adapt the requirements for bone tissue

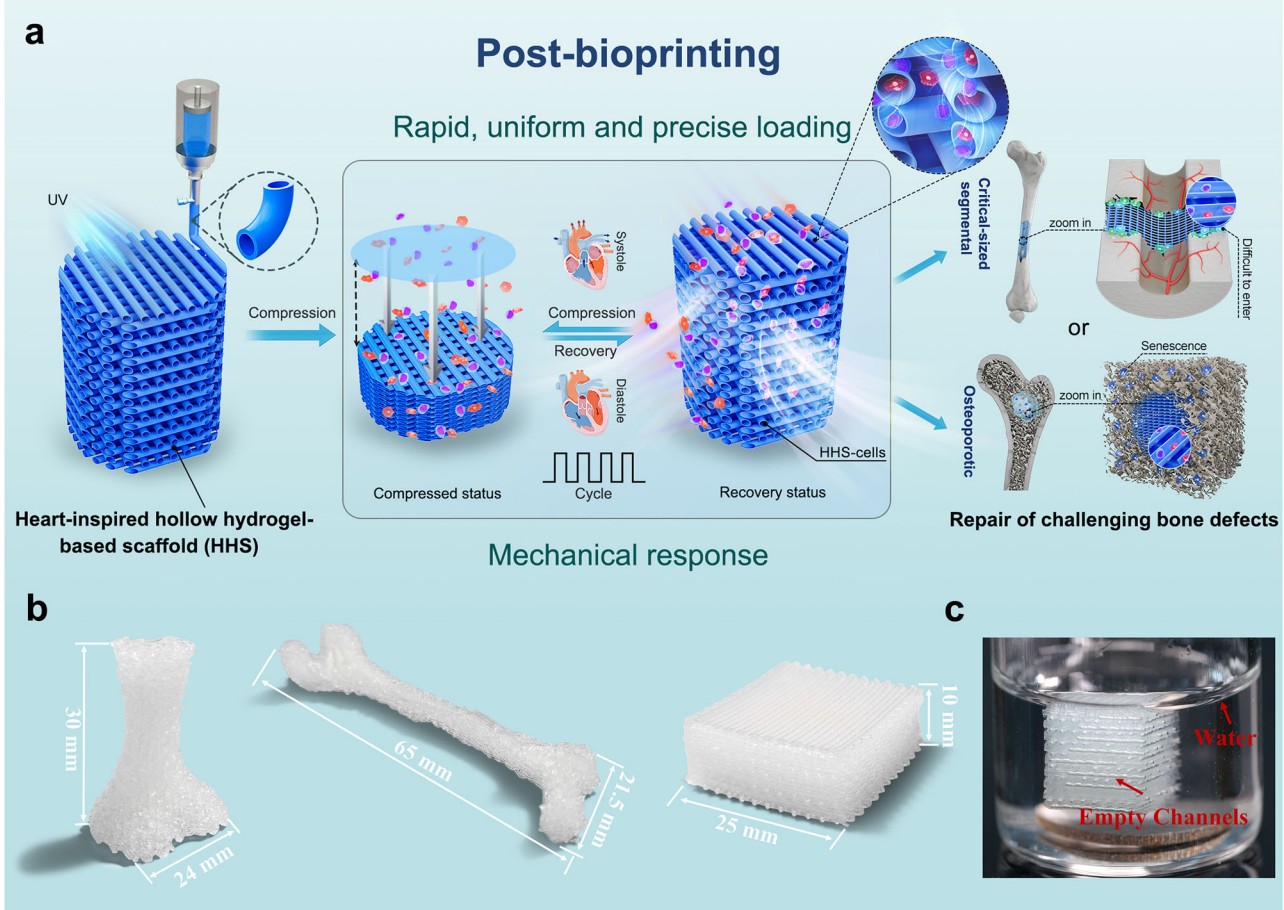

**Fig. 1 | Fabrication of large-sized and sophisticated HHSs. a** Schematic illustration of the mechanical-assisted post-bioprinting strategy and its application for repair of critical-sized segmental and osteoporotic bone defects. HHSs were fabricated by one-step coaxial printing without any supporting materials, HHSs could reversibly respond to an appropriate mechanical stimulation to rapidly, uniformly and precisely load cells without damage. **b** Photograph of the large-sized and bone-shaped HHSs with high shape fidelity. **c** Photograph of a HHS that suspended in water, the empty channels of the HHS were clearly observed.

engineering, variable mass ratios of N-acryloyl glycinamide (0, 4%, 8%, and 12%) hybrid hydrogel inks were primarily investigated, and noted as GLNx, (namely $x = 0$, 4, 8, and 12, respectively), while the other components were kept constant (gelatin methacryloyl, 12% w/v; Laponite nanoclay, 10% w/v). N-acryloyl glycinamide possessed multiple hydrogen bonding reinforcements and played a key role in upregulating the printability of GLN inks and mechanical properties of GLN hydrogels (Supplementary Fig. 3). As shown in SEM images (Supplementary Fig. 4), gelatin methacryloyl with relatively high molecular weights formed tailed bridges between nanoclay structures; while N-acryloyl glycinamide with low molecular weights could be physically adsorbed onto nanoclay surface, leading to a decrease of viscosity of GLN inks. Notably, compression modulus and strength of GLN12 could reach to $989 \pm 49$ KPa and $1200 \pm 78$ KPa, respectively (Supplementary Fig. 5a, b). Moreover, relaxation of stress occurred in all samples of GLN hydrogels (Supplementary Fig. 5c, d), suggesting the presence of abundant sacrificial bonds (primarily electrostatic interactions and multiple hydrogen bonds). Supplementary Fig. 5e, f displayed that GLN8 and GLN12 barely shrunk and exhibited slow degradation, could maintain their initial shape in PBS solution at 37 °C, which is very important for their corresponding HHSs to keep stable mechanical properties. Therefore, in light of its printability and mechanical outcomes, GLN12 should be the most suitable composition for fabrication of large-sized and sophisticated HHSs and was thus chosen as the basic ink for further assessments in this study.

## Tunable hollow structures and grids of HHSs

As presented in Fig. 2a, the inner and outer diameters ($d$ and $D$, respectively), the closest distance between two hollow filaments in a layer ($L$), the volume of hollow space of filaments in HHS ($V_1$), and the volume inside the grids ($V_2$), were defined. In an HHS, the substantial part was filled with the GLN hydrogel, while $V_1$ and $V_2$ could be representative for the space in between. As shown in Fig. 2b, c and Supplementary Fig. 6, the smooth and uniform hollow filaments, the integrity of hollow structure as well as the high fidelity of distinct grids could be clearly observed, and their $L$, $D$ and $d$ were tunable for regulation of HHS capability. $L_x$ ($x$ ranging from 0.2 to 0.8 mm) could be easily tuned by the mold design in the GeSiM Robotics Software, while $D_y$ ($y$ ranging from 0.4 to 0.8 mm), $d_z$ ($z$ ranging from 0 to 0.6 mm), and the wall thickness $D_y \cdot d_z$ (ranging from 0 to 0.4 mm) of hollow filaments could be regulated by adjusting the external and internal needle size of coaxial nozzles. Additionally, HHSs with different combinations of parameters were further displayed in Fig. 2c and their detailed analysis was summarized in Fig. 2d, suggesting that our printed HHSs possessed excellent designability.

## The compressibility, resilience, shape recovery, and fatigue resistance of HHSs

The effects of $d$ in hollow filaments on mechanical behaviors of HHSs ($LDd_z$) were comprehensively investigated by compression and cyclic compression tests. As shown in Fig. 3a(i) and Supplementary Fig. 7, the compression strength of HHSs improved with increase of $d$, whereas their compression modulus decreased; HHS ($L_{0.4}D_{0.6}d_{0.3}$) and HHS ($L_{0.4}D_{0.6}d_{0.4}$) could tolerate the compression up to 80% strain without fracture but HHS ($L_{0.4}D_{0.6}d_0$) and HHS ($L_{0.4}D_{0.6}d_{0.2}$) were fractured at ~47% and 65% strain, respectively. The resilience of HHSs ($L_{0.4}D_{0.6}d_{0.3}$ and $L_{0.4}D_{0.6}d_{0.4}$) was further demonstrated in Fig. 3a(ii) and Supplementary Movie 2 and these HHSs with large $d$ could recover to their initial states and remain intact with no damage even after 80% strain. Moreover, the resilient speed of HHS was visualized by luminescence in Fig. 3b and Supplementary Movie 3. The HHS ($L_{0.4}D_{0.6}d_{0.3}$) exhibited a significant deformation by a slight press, and immediately recovered to the original shape, meanwhile absorbing the luminescent liquid thoroughly after removal of press. In particular, it merely needed 4 s for one cycle of manual compression and shape recovery and the fast

recovery took place within only 1 s, similar with the cardiac impulse owing to systole and diastole. Subsequently, reusability and resistance to fatigue of HHSs were determined by a cyclic compression-recovery test at a strain of 40% (Fig. 3c, d). Results showed that stress was negatively correlated with $d$. Once undergoing $10^2$ cycles, the hysteresis area was dramatically reduced, indicating that severe damages occurred in the HHS ($L_{0.4}D_{0.6}d_0$); whereas the hysteresis area of the HHS ($L_{0.4}D_{0.6}d_{0.4}$) showed no significant decrease, and the HHSs ($L_{0.4}D_{0.6}d_{0.3}$ and $L_{0.4}D_{0.6}d_{0.4}$) could recover to its initial state and remain intact after even $10^4$ cycles. The results revealed that the anti-fatigue performance of HHSs could be enhanced by increasing $d$, especially the HHSs ($L_{0.4}D_{0.6}d_{0.3}$ and $L_{0.4}D_{0.6}d_{0.4}$) with an exceptional fatigue resistance (Fig. 3e). Taken together, owing to its hollow structure, HHS, possesses attractive compressibility and resilience, rapid shape self-recovery as well as exceptional fatigue resistance.

## Mechanical responsiveness of HHSs

On account of its particular mechanical features, HHS is suitable as a tissue-engineered scaffold for active uptake of bioactive substances (e.g., growth factors, cells) via an appropriate mechanical stimulation. In order to further explore their mechanical responsiveness, HHSs with tunable $d_z$ ($L_{0.4}D_{0.6}d_z$, $z = 0$, 0.2, 0.3, 0.4) and $L_x$ ($L_xD_{0.6}d_{0.4}$, $x = 0.2$, 0.4, 0.6) were elaborately fabricated. The mechanical responsiveness of HHSs was evaluated by immersing them in water with or without a mechanical stimulation. As shown in Fig. 4a, b, without mechanical stimulation, HHSs displayed a similar tendency to absorb water: the water uptake ratio of HHSs sharply raised within the initial 5 min and subsequently kept constant over time, showing a positive correlation with $L$ but remaining constant with $d$. In the absence of mechanical stimulation, the GLN hydrogel hardly took water over time, while the tendency of water uptake in HHS ($LDd_0$) was similar to other HHSs above with hollow structure, suggesting that their water uptake ability depends on the grids but not the swelling of GLN hydrogels or the hollow structure of HHSs (Fig. 4c). Water uptake ratios of HHSs were summarized in Supplementary Table 1, matching approximately the theoretical volume ratio of $V_2$. Taking HHS ($L_{0.2}D_{0.6}d_{0.4}$) as an example, $V_2$ ($27.7 \pm 5.4\%$) was close to the ratio of water uptake ($27.6 \pm 0.5\%$). Thus, it can be concluded that the space of $V_2$ is the main contributor to the water uptake in HHSs under a static condition.

Further, as shown in Fig. 4d–f, the mechanical-responsive ability of HHSs was determined under a dynamic condition. The water uptake ratio of HHSs exhibited a significantly increasing tendency with strain and number of cycles. In contrast, the water uptake ratio of HHS that had no $V_1$ kept constant with compressive strain. In addition, $L$ and $d$ of HHSs could also remarkably influence the water uptake: the water uptake ratio of $Lx$ HHSs tended to increase and then decrease with $L$, while the water uptake ratio of $d_z$ HHSs exhibited an increasing tendency with $d$. Theoretically, as $L$ increased, total number of the hollow filaments should be lowered resulting in a decrease of theoretical volume of $V_1$ (Supplementary Table 1), and thus the theoretical volume ratio of $V_1$ in $L_{0.6}$ HHS ($18.3 \pm 2.8\%$) was the smallest. Despite the biggest theoretical volume ratio of $V_1$ in $L_{0.2}$ HHS ($28.7 \pm 2.5\%$), the theoretical volume ratio of $V_2$ in $L_{0.2}$ HHS (only $27.7 \pm 5.3\%$), smaller than that of $V_1$, could not provide enough space to support the deformation of $V_1$, resulting in a restraint of water uptake in $L_{0.2}$ HHS. Thus, the results reasonably displayed that the water uptake ratio in $L_{0.4}$ HHS was higher than those of $L_{0.2}$ and $L_{0.6}$ HHSs (Fig. 4d). In addition, the water uptake ratio of $d_z$ HHSs exhibited an increasing tendency with $d$ (Fig. 4e). In particular, by comparing water uptake ratios and theoretical volume ratios of $V_1$ in $d_z$ HHSs, it was found that $V_1$ of HHSs could be completely full with water at a strain of 80%. For example, the water uptake ratio in $d_{0.4}$ HHS reached to $23.3 \pm 6.8\%$, very close to the theoretical volume ratio of $V_1$ in $d_{0.4}$ HHSs ($22.1 \pm 3.3\%$) (Fig. 4e and Supplementary Table 1). In light of the water uptake ability, a HHS ($L_{0.4}D_{0.6}d_{0.4}$) was chosen for further assessment of multicyclic strain loading (Fig. 4f) and

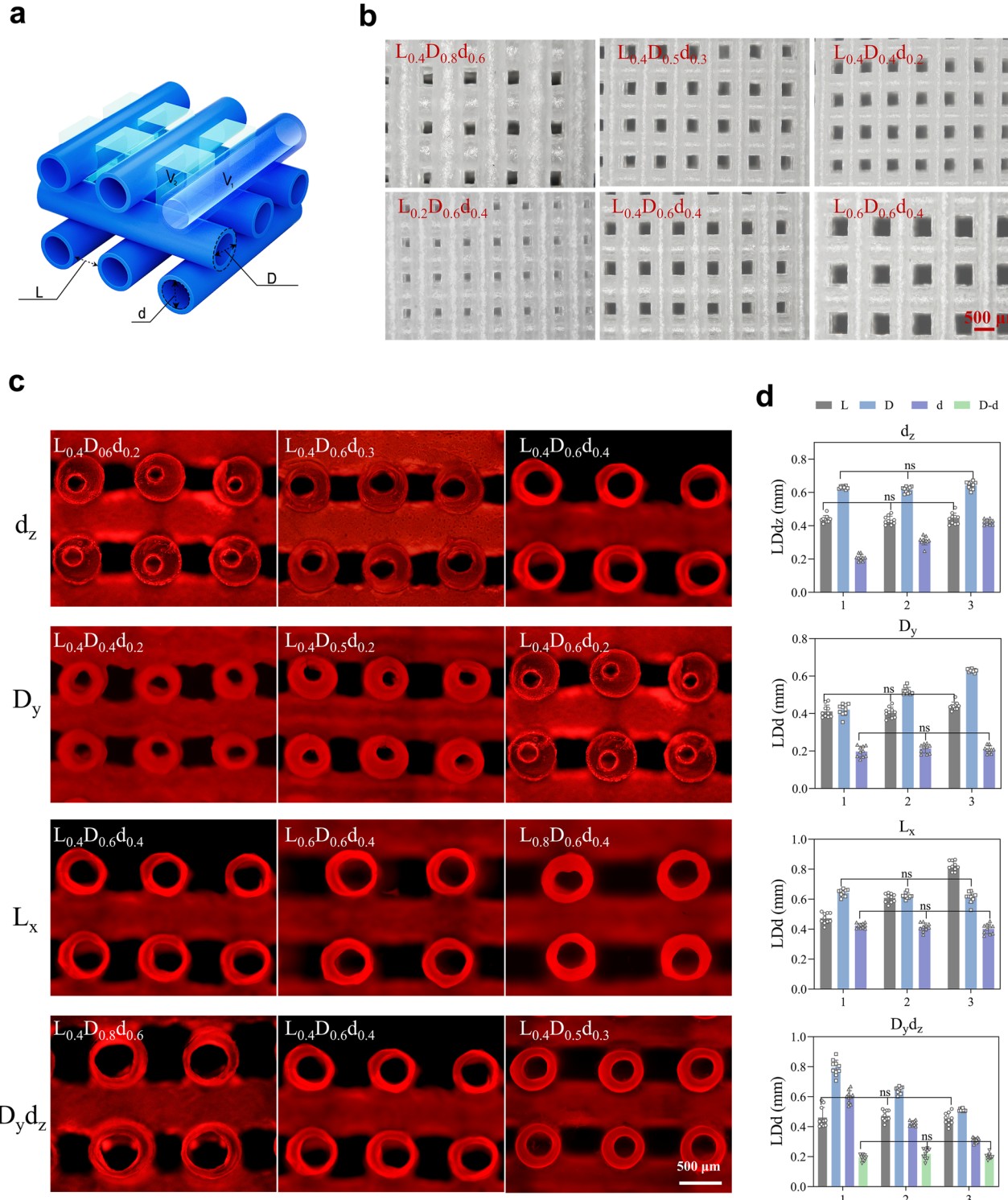

**Fig. 2 | Tunable hollow structures and grids of HHSs. a** Illustration of the critical parameters of HHSs. **b** Gross morphology of the filaments and grids of HHSs. **c** Fluorescent images of the hollow structures of HHSs. Tunable hollow structures and grids were controlled by $L$, $D$, and $d$. **d** Quantitative analysis of the HHSs with different combinations of parameters: same $L$ and $D$ but different $d$, same $L$ and $d$ but different $D$, same $D$ and $d$ but different $L$, and same $L$ and $D - d$ but different $D$ and $d$. $n = 10$ samples. Data are presented as mean ± s.d, statistical significance was calculated using two-way ANOVA method with Tukey's multiple comparisons tests, ns represents no significant difference ($P > 0.05$). Source data are provided as a Source Data file.

the results showed that the water uptake ratio significantly elevated with loading cycles under both 20% and 40% strain. Thus, it can be concluded that $V_1$ contributes to the improvement of water uptake in the HHSs under a dynamic condition, providing HHSs with excellent mechanical-responsive ability.

To further visualize the mechanical responsiveness, HHSs ($L_{0.4}D_{0.6}d_z$, z = 0, 0.2, 0.4, 0.6) were immersed in a fluorescent microsphere solution. Single cycles with multiple strains (0, 20%, 40%, 60%, and 80%) and multi-cycles (from 0 to 10) at 20% or 40% strain on the HHSs were demonstrated. The fluorescence images were acquired via

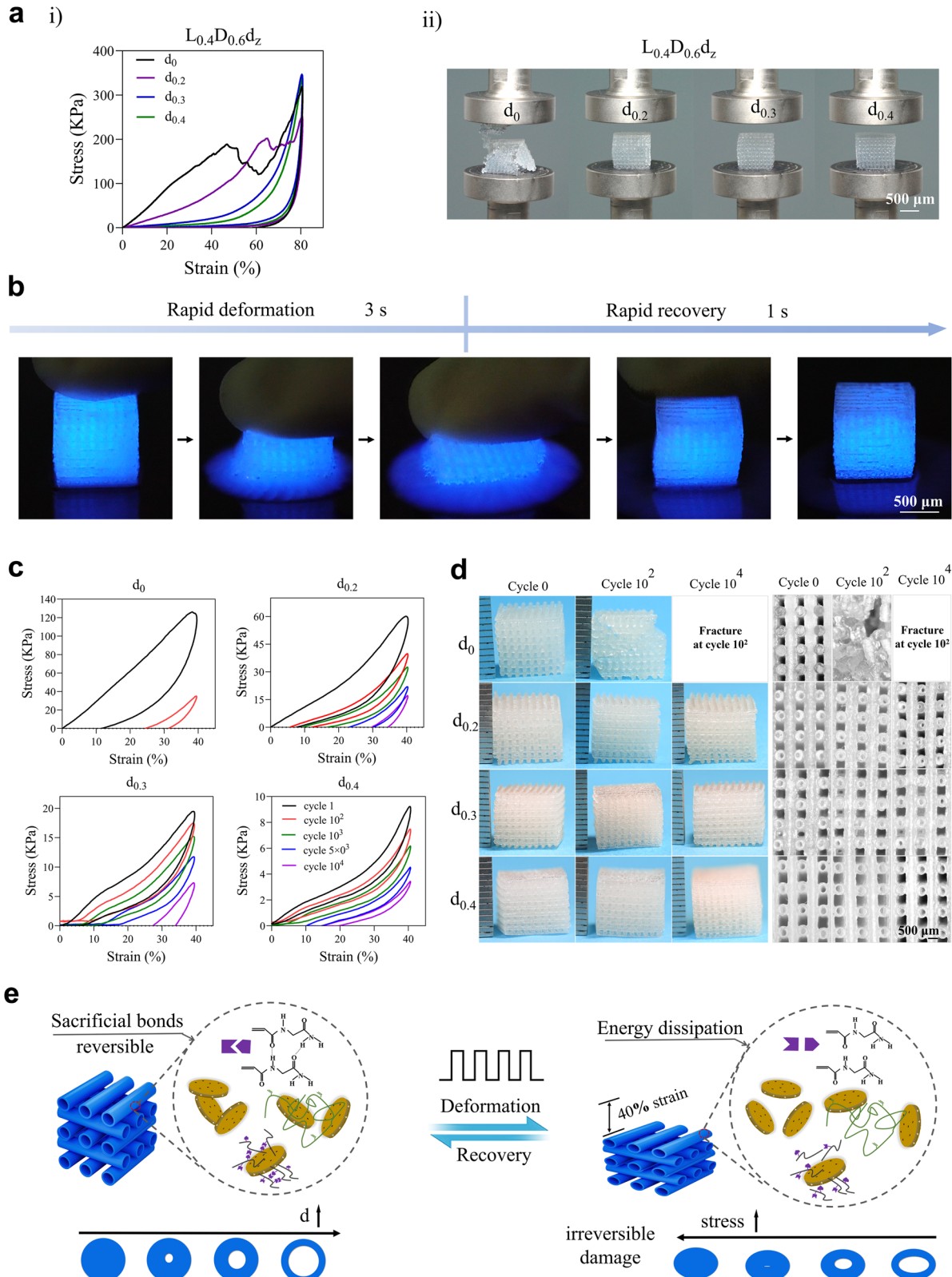

**Fig. 3 | Mechanical properties of HHSs ($L_{0.4}D_{0.6}d_z$). a** Compression test of HHSs. (**i**) Compression-recovery curve with 80% strain of HHSs; (**ii**) Photograph of HHSs after removal of compression with 80% strain. HHSs ($L_{0.4}D_{0.6}d_0$ and $L_{0.4}D_{0.6}d_{0.2}$) were fractured with 80% compression strain, while HHSs ($L_{0.4}D_{0.6}d_{0.3}$ and $L_{0.4}D_{0.6}d_{0.4}$) could recover to their initial state and remain intact without fracture. **b** Photographs of the process of an HHS ($L_{0.4}D_{0.6}d_{0.3}$) absorbing luminescent solution from deformation to recovery, which was similar with the systole and diastole. Cyclic stress-strain curves of HHSs (**c**), and gross morphology and hollow structures of HHSs after 0, $10^2$, $10^3$, and $10^4$ cycles (**d**) at a strain of 40%. **e** Illustration of mechanism of HHSs for resistance to fatigue. The sacrificial bonds in the backbone of HHSs, including electrostatic interactions and multiple hydrogen bonds, were mostly reversible, and ruptured to dissipate energy during deformation and recovered after relaxation. Source data are provided as a Source Data file.

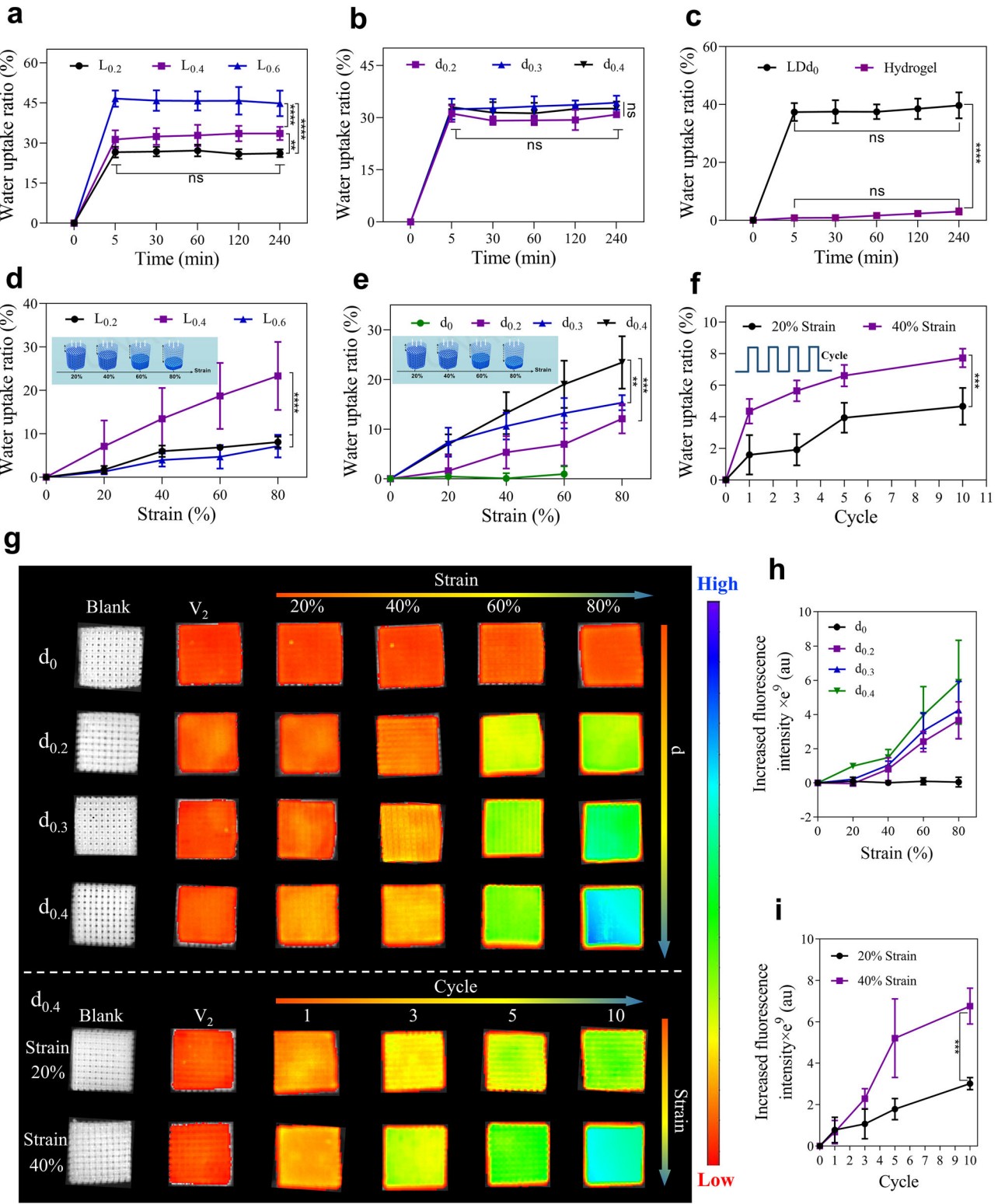

**Fig. 4 | Mechanical responsiveness of HHSs.** Water uptake of HHSs with various L ($L_xD_{0.6}d_{0.4}$, $x$ = 0.2, 0.4, 0.6), $n$ = 4, **$P$ = 0.006, ***$P$ = 0.002, ****$P$ < 0.0001, ns represents no significant difference ($P$ > 0.9999) (**a**) and $d$ ($L_{0.4}D_{0.6}d_z$, $z$ = 0.2, 0.3, 0.4), $n$ = 3, $P$ = 0.8302 ($d_{0.2}$ and $d_{0.3}$), $P$ = 0.997 ($d_{0.2}$ and $d_{0.4}$), $P$ = 0.9999 ($d_{0.3}$ and $d_{0.4}$) at 240 min, $P$ > 0.9999 at 5 min and 240 min of $d_{0.2}$, $d_{0.2}$, and $d_{0.3}$ (**b**), and GLN hydrogel and the HHS ($L_{0.4}D_{0.6}d_0$), $n$ = 3, ****$P$ < 0.0001, ns represents no significant difference ($P$ > 0.9999) (**c**) over time without mechanical stimulation under a static condition. Water uptake of HHSs with various L ($L_xD_{0.6}d_{0.4}$, $x$ = 0.2, 0.4, 0.6), $n$ = 4, ****$P$ < 0.0001 (**d**) and $d$ ($L_{0.4}D_{0.6}d_z$, $z$ = 0, 0.2, 0.3, 0.4), $n$ = 3, **$P$ = 0.0098, ***$P$ = 0.0003 (**e**) with increased compression stains (20%, 40%, 60%, and 80%) under a dynamic condition. **f** Water uptake of the HHS ($L_{0.4}D_{0.6}d_{0.4}$) subject to incremental numbers of cycles of compression-recovery (1, 3, 5, and 10) with 20% and 40% compression strains, $n$ = 3, ***$P$ = 0.0004. Visual fluorescence images acquired via an IVIS systems (**g**) and quantitative analysis of fluorescence intensity of the HHS ($L_{0.4}D_{0.6}d_z$, $z$ = 0, 0.2, 0.3, 0.4) subject to different strains from 0 to 80% at one cycle, $n$ = 3 (**h**) and cycles from 0 to 10 at 20% or 40% strain, $n$ = 3, ***$P$ = 0.0003 (**i**) to load the fluorescence microspheres. Colors here represent gradients in fluorescence intensity (red: low, blue: high), Ex = 605 nm, Em = 680 nm. Data are presented as mean ± s.d, statistical significance was calculated using two-way ANOVA method with Tukey's multiple comparisons tests. Source data are provided as a Source Data file.

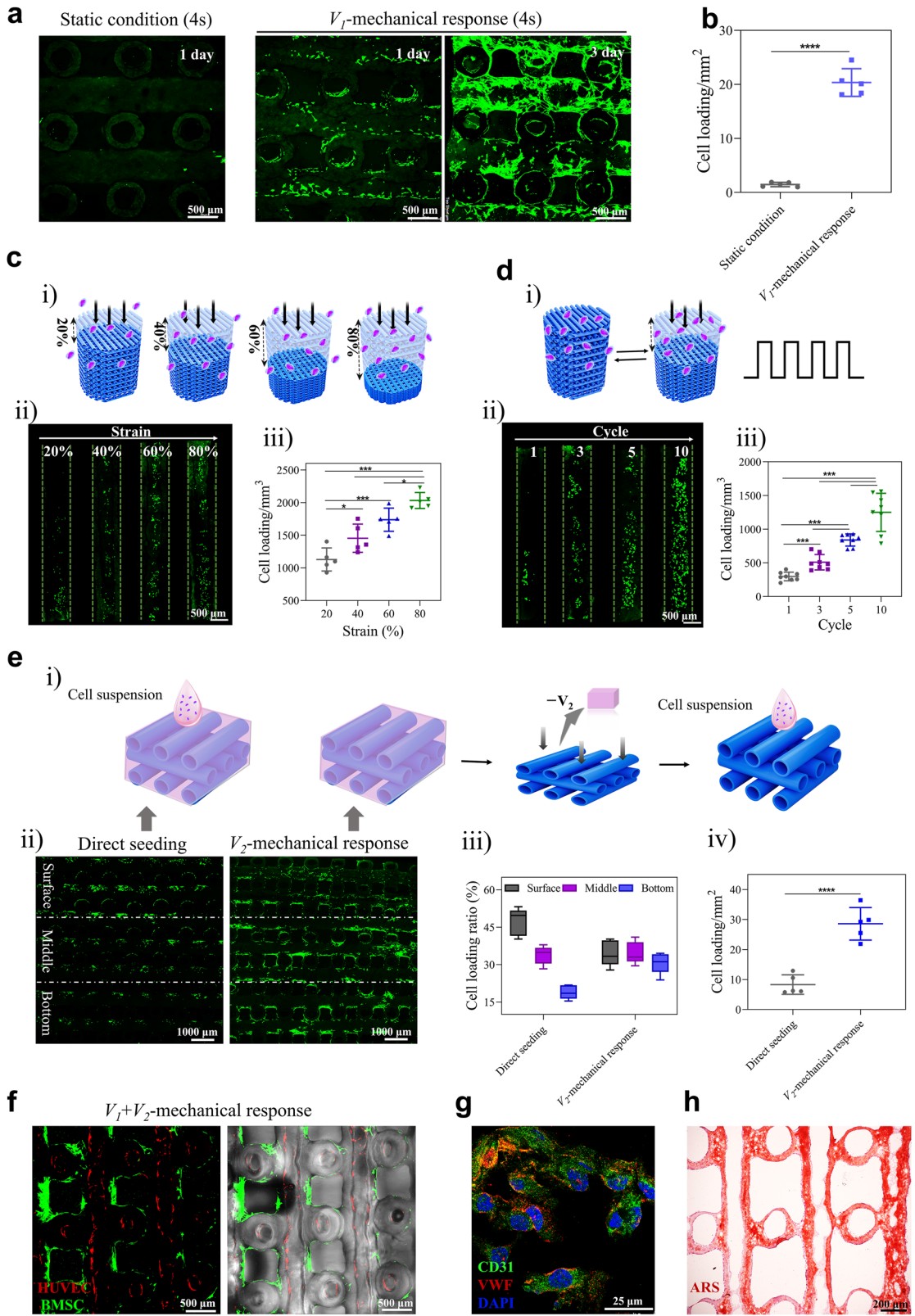

an IVIS system as shown in Fig. 4g. The fluorescent intensity was depicted by color gradients (red: low, blue: high), and the results from fluorescent intensity in the HHSs were consistent with those of described in Fig. 4b, e, f, further confirming that HHSs can rapidly respond to external mechanical stimulation and their mechanical-responsive ability can be enhanced with an increase in $d$, strain as well as number of cycles.

## Rapid, uniform, and precise cell loading capability of HHSs

In this study, a mechanical-assisted post-bioprinting strategy was developed to realize rapid, uniform, and precise cell loading on basis of the rapid compression-recovery ability and excellent mechanical responsiveness of HHSs. Human bone marrow mesenchymal stem cells (hBMSCs) are used as the cell model for these evaluations and the cell-laden constructs will be abbreviated to HHSs-cells in the following

**Fig. 5 | Cell loading to HHSs with mechanical responsiveness.** Fluorescence images (**a**) and quantitative analysis (**b**) of loaded cells in HHSs ($L_{0.4}D_{0.6}d_{0.4}$), $n = 5$, ****$P < 0.0001$. HHSs were subject to a static condition without mechanical stimulation within 4 s in cell suspension ($1 \times 10^6$ cells/mL), or a cycle of compression-recovery within 4 s at 80% strain in hBMSCs cell suspension ($1 \times 10^6$ cells/mL) under a dynamic condition. Fluorescence images of fluorescein diacetate staining and quantitative analysis were acquired at 1 and 3 days after HHS-cells culture in medium. **c** Schematic diagram (**i**), fluorescence images (**ii**) and quantitative analysis (**iii**) of cell loading into HHSs in hBMSCs cell suspension ($1 \times 10^6$ cells/mL), which were controlled by various compression strains (20%, 40%, 60% and 80%), $n = 5$, *$P = 0.0446$ (20% and 40%), ***$P = 0.0003$ (20% and 60%), ***$P = 0.0005$ (40% and 80%), *$P = 0.0156$ (60% and 80%). **d** Schematic diagram (**i**), fluorescence images (**ii**) and quantitative analysis (**iii**) of cell loading into HHSs in hBMSCs cell suspension ($5 \times 10^6$ cells/mL), which were controlled by number of cycles (1, 3, 5, and 10), $n = 8$, ***$P < 0.001$. **e** Schematic diagram (**i**), fluorescence images (**ii**), and quantitative analysis of cell distribution (**iii**) and cell numbers (**iv**) by direct seeding and $V_2$-mechanical response, respectively, $n = 5$, ****$P < 0.0001$. **f** Fluorescence images of HUVECs-RFP (red) and hBMSC-GFP (green) that were precisely partitioned in the hollow filaments and the grids of HHSs by successive combination of $V_1 + V_2$-mechanical response. **g** Protein secretion of CD31 (green) and VWF (red) of HUVECs in an HHS after culturing for 14 days. **h** Alizarin Red S staining (ARS) in an HHS with hBMSCs at day 21. Each experiment in (**f**–**h**) was repeated three times independently with similar results. Dates are presented as means ± s.d, statistical significance was calculated using Student's $t$ tests (**b**, **e** (**iv**)) and one-way ANOVA method with Tukey's multiple comparisons tests (**c** (**iii**), **d** (**iii**)). Source data are provided as a Source Data file.

context and As shown in Fig. 5a, b, compared with static conditions, cell numbers in HHSs under a cycle of compression-recovery within 4 s at 80% strain significantly improved about 13 folds, suggesting the activeness and effectiveness of our proposed strategy via the $V_1$-mechanical response. After 3 days of culture, the loaded cells in HHSs exhibited uniform deposition and obvious proliferation. Meanwhile, the number of loaded cells in HHSs increased with compression strains (Fig. 5c) or number of cycles (Fig. 5d), suggesting the controllability of our proposed strategy. In addition, the larger size of HHSs ($15 \times 15 \times 10$ mm and $20 \times 10 \times 10$ mm) could also rapidly load cells under a cycle of compression-recovery within 4 s at 80% strain (Supplementary Fig. 9). Furthermore, mitochondrial membrane potential, reactive oxygen species (ROS) and proliferation of the cells were examined after mechanical stimulation (80% strain or 10 cycles of compression-recovery, respectively), the results proved that the mechanical stimulation has no negative influences on viability of cells (Supplementary Fig. 10). Accordingly, this post-bioprinting strategy for loading cells into HHSs was cell-friendly and convenient.

Partitioned introduction of multiple cell types into scaffolds is often required but still unmet in traditional tissue engineering. In this study, taking advantage of different mechanical responsiveness between $V_1$ and $V_2$, two types of cells (hBMSCs and human umbilical vein endothelial cells (HUVECs)) were precisely loaded into $V_1$ or $V_2$, respectively. As shown in Fig. 5e(i), a simple approach of $V_2$-mechanical response was further developed to remove medium only in $V_2$, subsequently seeding hBMSCs suspension, thereby regulating cell number and uniformity. After 4 h of culture, hBMSCs could adhere in the HHS owing to good adhesive properties of GLN hydrogel (Supplementary Fig. 8). It was clearly observed that hBMSCs distributed uniformly in the whole $V_2$ of HHSs (surface-middle-bottom) (Fig. 5e(ii), e(iii)) and the number of hBMSCs loaded in $V_2$ of HHSs could be improved about 200% (Fig. 5e(iv)) via the $V_2$-mechanical response compared with direct cell seeding, suggesting the availability of $V_2$ in HHSs. The processes of direct cell seeding and $V_2$-mechanical response approaches were displayed in Supplementary Movie 4. As shown in Fig. 5f and Supplementary Movie 5, by successive processes of loading HUVECs-RFP via $V_1$-mechanical response and seeding hBMSCs-GFP through $V_2$-mechanical response ($V_1 + V_2$-mechanical response), HUVECs-RFP and hBMSCs-GFP were distributed in $V_1$ and $V_2$, respectively. Furthermore, immunofluorescence staining of CD31 and VWF proved that the endothelial phenotype could be maintained and intercellular junctions between HUVECs preliminarily formed[29] in HHSs after 14 days of culture (Fig. 5g). Additionally, HHSs loaded with hBMSCs were cultured in osteogenic differentiation media. Together with enhancements in proliferation, osteogenic protein secretion, and gene expression of hBMSCs in GLN hydrogels (Supplementary Fig. 11), significant calcium deposition was also observed in HHSs at 21 days of culture (Fig. 5h). The results indicated that this post-bioprinting strategy could realize precisely partitioned loading of multiple cells and HHSs could promote osteogenesis and angiogenesis in vitro.

## In vivo repair of challenging bone defects by HHSs-cells

The capability of HHS-cells for bone defect repair was assessed in critical-sized segmental and osteoporotic bone defect models in *rats*. Firstly, inflammation and foreign body response of HHS-M (HHS with rBMSCs) were evaluated through subcutaneous implantation in rats. In comparison to HHS without cells, the results of indicated that HHS-M did not display excessive inflammation and foreign body response (Supplementary Fig. 12). Bioluminescence images and a quantitative analysis of HHS-M with luciferase-overexpressed rBMSCs in vivo (Supplementary Fig. 13), showed that the majority of cells seeded in the HHS could survive, although a portion of the seeded cells died. The schematic diagram of the experiments was shown in Fig. 6a and the surgical procedure was demonstrated in Supplementary Fig. 14 From X-ray images in Fig. 6b, no bridging was observed in Blank, HHS, HHS-M (HHS with rBMSCs), and HHS-ME (HHS with rBMSCs and rECs) groups at 6 weeks, while a complete bridging in HHS-M and HHS-ME was observed at 12 weeks. As shown in Fig. 6c, the morphological characteristics of newly formed vessels and bones were visually mirrored by 3D reconstruction from μCT images. At 6 weeks postoperatively, the results from microfil perfusion showed that there were more newly and orderly formed vessels in the HHS-cells (especially in the HHS-ME) than those of HHSs and Blank (local magnifications in Supplementary Fig. 15). At 12 weeks postoperatively, HHS-M and HHS-ME groups exhibited plenty of new bone formation, and tubular and rod-like structures were clearly observed to bridge the proximal and distal of femoral defect, indicating that new bone can form inside HHS-M and HHS-ME (Fig. 6c). Quantification of bone volume (BV), bone trabeculae thickness (Tb.Th) and bone trabeculae number (Tb.N) within the defect from the reconstructed μCT images were summarized in Fig. 6d. The significant increase of BV (Fig. 6d) in 4 groups from 6 to 12 weeks was seen. At 12 weeks, compared to Blank and HHS, both HHS-M and HHS-ME supported statistically greater levels of BV, and the BVs of HHS-M and HHS-ME groups were 2.1- and 2.5-fold higher than that of the HHS group, respectively. A significant increase of Tb.Th was also found in both HHS-M and HHS-ME groups compared to Blank and HHS. There was no difference in Tb.N between 4 groups. As shown in Fig. 6e, Hematoxylin and eosin (H&E) and Masson's trichrome staining further confirmed that the HHSs-cells (HHS-M and HHS-ME) significantly improved the reconstruction of new bone, and HHS-ME seemed to be slightly better than HHS-M in promoting bone regeneration.

As shown in Supplementary Fig. 16a, b, *rat* osteoporotic model was established successfully. More importantly, cellular senescence-associated βgalactosidase (SAβgal) activity and senescence-related gene expression of p16 both increased in BMSCs derived from osteoporotic *rats* (OVX-BMSCs) compared to BMSCs derived from healthy and young *rats* (Supplementary Fig. 16c, d). In contrast, the proliferation behavior decreased in OVX-BMSCs (Supplementary Fig. 16e). Additionally, the ability of osteogenic differentiation in OVX-BMSCs significantly diminished (Supplementary Fig. 16f, g). The results confirmed that the OVX-BMSCs in the osteoporotic *rats* were senescent.

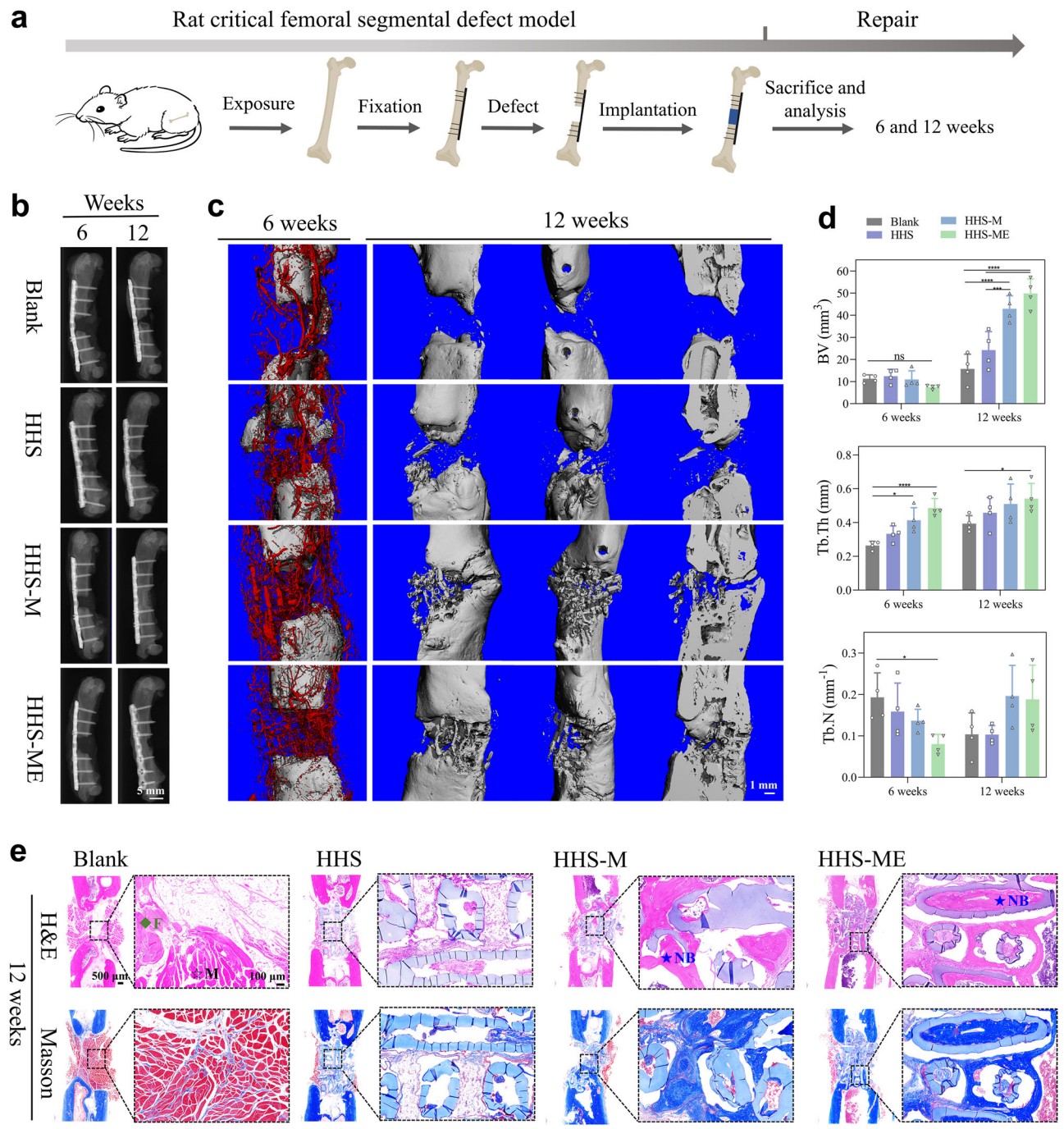

**Fig. 6 | Reconstruction of large-sized segmental bone defects in *rats* by Blank, pure HHS, and HHS-cells (HHS-M and HHS-ME). a** Schematic diagram of the experiments. **b** Radiography (X-ray) assessment at week 6 and 12 after surgery. **c** Microcomputed tomography (μCT) reconstruction images of bone at week 6 and 12, and blood vessels (red) at week 6 after surgery. **d** Quantitative analysis of bone volume (BV), trabecular number (Tb.N), and trabecular thickness (Tb.Th) from the reconstructed μCT images at week 6 and 12. Data are presented as means ± s.d, $n = 4$ per group, statistical significance was calculated using one-way ANOVA method with Tukey's multiple comparisons tests, BV ***$P = 0.0008$ and ****$P < 0.0001$, ns represents no significant difference (P å 0.05), Tb. Th *$P = 0.0103$ and ***$P = 0.0008$ at week 6 and *$P = 0.0271$ at week 12, Tb. N *$P = 0.0121$. **e** Histological staining (H&E and Masson's trichrome) at week 12. A visible defect gap in the Blank and pure HHS groups, positive staining of new bone in the HHS-M and HHS-ME groups, were observed according to H&E (pink) and Masson's trichrome (deep blue) staining, each experiment was repeated four times independently with similar results. Magnified images are in the dotted boxes. Green rhombus, black hollow five-pointed star, and blue solid five-pointed star represent fibrous, muscle, and new bone tissues, respectively, Source data are provided as a Source Data file.

To further investigate the bone regenerative ability of HHS-cells under osteoporotic conditions, cylindrically shaped scaffolds were implanted into metaphyseal defects in osteoporotic rat femurs (Supplementary Fig. 17) and the schematic diagram was shown in Fig. 7a. After 4 and 8 weeks of surgery, most new bone formation was observed in HHS-M group, and the Blank group exhibited the worst effects judging from the visibly empty area inside even at 8 weeks (Fig. 7b, d). New bone formation after 4 and 8 weeks of surgery were further visualized by μCT, as shown in Fig. 7b. Notably, a large amount of new bone formation could be seen at the region of cancellous bone defect in the HHS-M group, while in the Blank group almost no signal of new bone was revealed even at 8 weeks postoperatively. Quantification of ratio of

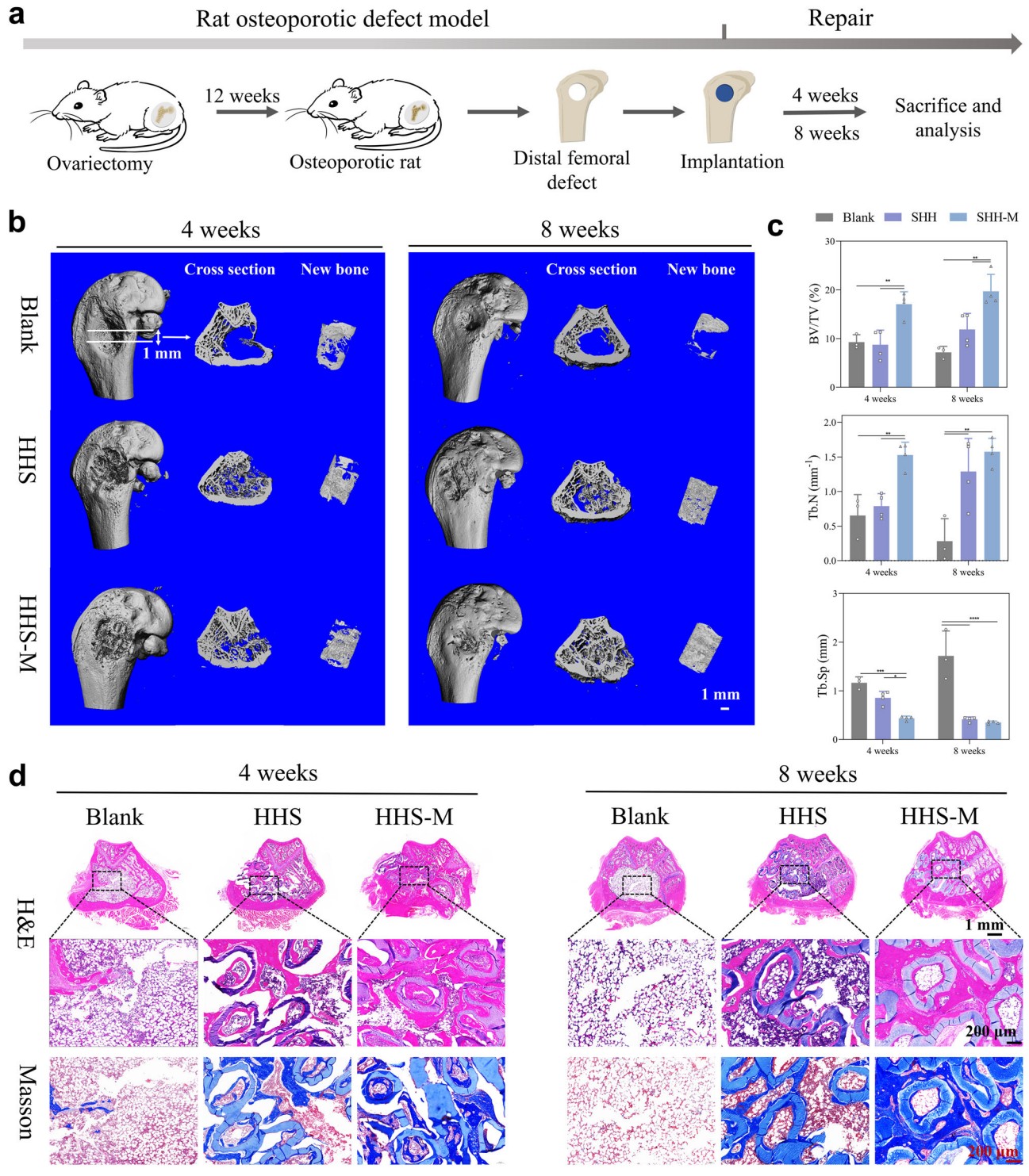

**Fig. 7 | Regeneration of osteoporotic bone defects in *rats* by Blank, pure HHS and HHS-cells (HHS-M).** **a** Schematic diagram of the experiments of bone healing in the distal femoral metaphyseal critical-size defects of osteoporotic *rats*. **b** μCT reconstruction images showing overall views of the bone specimens, cross-section views and new bone views in the defective region at week 4 and 8 after surgery. **c** Quantitative analysis of ratio of bone volume to the total defect volume (BV/TV), bone trabeculae number (Tb.N), and trabecular separation (Tb.Sp) from the reconstructed μCT images at week 4 and 8. Data are presented as means ± s.d, *n* = 3 for Blank groups, *n* = 4 for HHS and HHS-M, statistical significance was calculated using two-way ANOVA method with Tukey's multiple comparisons tests, BV/TV of Blank and HHS-M **$P$ = 0.006 (week 4) and ****$P$ < 0.0001 (week 8), BV/TV of HHS and HHS-M **$P$ = 0.0018 (week 4) and **$P$ = 0.003 (week 8), Tb.N of Blank and HHS-M **$P$ = 0.0036 (week 4) and ****$P$ < 0.0001 (week 8), Tb.N **$P$ = 0.0072 (HHS and HHS-M at week 4) and **$P$ = 0.0011 (Blank and HHS at week 8), Tb.Sp at week 4 ***$P$ = 0.0005 (Blank and HHS-M) and *$P$ = 0.0209 (HHS and HHS-M), Tb.Sp at week 8 ****$P$ < 0.0001. **d** H&E and Masson's trichrome staining at week 4 and 8, each experiment was repeated four times independently with similar results. Magnified images are in the dotted boxes. A visibly empty area was spotted in Blank group even at week 8, obvious new bone formation was seen in HHS-M groups at week 4 and 8. Source data are provided as a Source Data file.

bone volume to the total defect volume (BV/TV), Tb.N and trabecular separation (Tb.Sp) within the defect from the reconstructed μCT images were summarized in Fig. 7c. The results from BV/TV and Tb.N revealed that the levels of HHS-M group were always higher than those of HHS and Blank group at all time points, with an order of HHS-M > HHS > Blank at 8 weeks. A significant decrease of Tb.Sp was found in HHS and HHS-M group, among which Tb.Sp in the HHS-M group was the lowest. The results from H&E and Masson's trichrome staining (Fig. 7d) showed that HHS-M groups showed the most new bone formation at both 4 and 8 weeks, which is consistent with the results from μCT images. Although the GLN hydrogels demonstrated certain bioactivity to promote blood vessels ingrowth and bone regeneration (Supplementary Figs. 11, 18), the weak effects of pure HHSs on bone formation were observed under osteoporotic condition. In contrast, massive new bone was detected on the inner and outer walls of the hollow tubes in the HHS-M group, suggesting the necessity of loading healthy cells in osteoporotic bone defects. Accordingly, it can be concluded that MSCs-laden HHSs can accelerate bone repair, offering a promising strategy to treat osteoporotic defects.

## Discussion

The bioprinting technology has revolutionized bone tissue engineering to become more time-saving and effective, e.g., it could realize readily a desired level of cell deposition and distribution in scaffolds[16,17,21,22]. However, it remains challenge to maintain cell viability during the bioprinting process and ameliorate mechanical stability of bioprinted cell-laden constructs, which is a requirement for bone regeneration[24–27]. Especially for extrusion bioprinting, cells are partially damaged by shear stress during extrusion and complex structures often fail due to unsatisfying printability and poor self-supporting ability of bioinks[26], which severely hinders the application for challenging bone defect repair[30,31]. Therefore, it encourages us to present a post-bioprinting strategy based on the rapid compression-recovery. Compared to other cell loading methods previously reported[7,16,23,32–34] in terms of time, uniformity, condition, designability, and size (Supplementary Fig. 19), excellent mechanical responsiveness of HHSs can realize rapid, uniform, friendly, and precise cell loading for fabrication of large-scale cell-laden constructs, strengthening the tissue-engineered therapy for repair of large-sized segmental and osteoporotic bone defects.

For this post-bioprinting strategy, 3D printing of constructs with hollow structures cannot be overemphasized. Sacrificial templates are commonly used in combination with 3D printing to fabricate constructs with hollow structures, and disposal of the supporting material is usually necessary[35,36]. Besides, low concentrations of hydrogel bioink in extrusion bioprinting often lead to insufficient mechanical stability and printability, making it difficult to bio-print constructs with hollow structures with an adequate structural integrity[23]. Herein, for the first time, large-sized HHSs were precisely fabricated through a simple and one-step process of coaxial printing technique without any supporting materials. Remarkably, HHSs with a 6 cm height were successfully obtained, almost reaching the critical size for human segmental bone defect[37]. The superior printability, rapid shape-recovery behaviors, and excellent self-supporting ability of GLN hydrogel inks are the critical prerequisite for successful fabrication of HHSs, attributable to their abundant physically interpenetrated networks. On one hand, the anisotropic charge distribution on the nanoclay (positive edges, negative faces) results in formation of a "house-of-cards" structure, contributing to the shear-thinning properties of nanoclay-polymer solutions[38]. Besides, backbones of gelatin methacryloyl and N-acryloyl glycinamide interact with charged surface/edge of nanoclay via electrostatic interactions. gelatin methacryloyl with relatively high molecular weights, tails as a bridge between multiple nanoclay resulting in the formation of a physically crosslinked network and sharply increase the viscosity of the hybrid inks. The N-acryloyl glycinamide also affects the interaction between gelatin methacryloyl and nanoclay, causing a reduction in the viscosity of GLN inks. One the other hand, reversible multiple hydrogen bonding also occurs among N-acryloyl glycinamide molecules[30,31]. Specifically, the electrostatic interactions and multiple hydrogen bonds will break and reform reversibly once applying mechanical stress and relaxation to the system[38]. Besides printability and self-supporting of GLN inks, structural stability and sufficient mechanical strength under physiological conditions[39] are also critical for achievement of the tunable HHSs with high fidelity. Structure stability of HHS is affected by solidification processes and swelling properties of GLN hydrogels[25]. The covalent bonds formed in HHSs by photo-crosslinking can maintain the stability and integrity of HHSs under physiological conditions without collapse. Besides, the introduction of N-acryloyl glycinamide contents can improve hydrogen bonds and increase covalent crosslinks between N-acryloyl glycinamide and gelatin methacryloyl chains, restricting the diffusion of water molecules into or out of the GLN hydrogel network, thus offering sufficient mechanical strength.

Owing to the tunable hollow structure of GLN hydrogels, HHSs with attractive resilience, rapid shape recovery, and exceptional fatigue resistance are thoroughly demonstrated, which are the keys to success of this mechanical-assisted post-bioprinting strategy for loading cells. The substantial part of GLN hydrogel, and space parts of $V_1$ and $V_2$, together regulate the mechanical properties of the HHSs. Materials with excellent resilience can perform a large overall deformation with little energy dissipation[40]. However, as hydrogels are viscoelastic, large deformation can generally convert into pronounced energy dissipation. In this study, low hysteresis and large deformation were achieved attributing to the existence of $V_1$ in the HHSs. $V_1$ provided the space to significantly deform (80% strain) for the backbone (GLN hydrogel) of HHSs. Besides, the air in $V_1$ could be expelled when compression occurred, resulting in lower pressure. Subsequently, the pressure difference between the inside and outside of HHSs endowed them with rapid recover ability under one cycle of compression-recovery process within 4 s, demonstrating that $V_1$ caused the aforementioned mechanical responsiveness. The compression-recovery process of HHS is similar with the systole-diastole of heart. The dissipated energy in the compression-recovery cycles was linked to the rupture of the physical crosslinks[41]. The sacrificial bonds, such as "house-of-cards" structure of nanoclay[38], electrostatic interactions among nanoclay-polymer chains (gelatin methacryloyl or NAGAN-acryloyl glycinamide)[30,38], and multiple hydrogen bonding among NAGAN-acryloyl glycinamide molecules[31], responsible for energy dissipation, existed in the backbone (GLN hydrogel) of HHSs. Notably, as depicted in Fig. 3e, sacrificial bonds in the HHSs were mostly reversible, and ruptured to dissipate energy during material deformation and recovered after relaxation[42]. In detail, however, they would begin to irreversibly break with large deformation that dissipated excessive energy. Minimization of energy dissipation was essential for achieving high performance of the HHSs where repetitive actions were required. The solid content of HHSs would decrease with an $V_1$, resulting in production of relatively lower stress on the HHSs ($L_{0.4}D_{0.6}d_{0.3}$ and $L_{0.4}D_{0.6}d_{0.4}$) under a same deformation. Thus, after $10^4$ cycles, only slightly irreversible damage of sacrificial bonds occurred in the HHSs ($L_{0.4}D_{0.6}d_{0.3}$ and $L_{0.4}D_{0.6}d_{0.4}$), demonstrating their exceptional fatigue resistances. However, continuously high stress would cause irreversible damage of sacrificial bonds in the HHSs ($L_{0.4}D_{0.6}d_0$). The energy dissipation in the HHSs ($L_{0.4}D_{0.6}d_0$) without $V_1$ was insufficient to resist cycles of continuously high stress, and caused irreversible damage of sacrificial bonds, leading to severe damage of the HHSs after $10^2$ cycles. In short, the existence of $V_1$ significantly elevated the reusability and resistance to fatigue of HHSs.

More importantly, by means of this post-bioprinting strategy, HHSs-cells are successfully achieved, attributing to distinct mechanical responsiveness between $V_1$ and $V_2$. HHSs with $V_1$ presented attractive resilience, could undergo rapid compression-recovery within 4 s under a dynamic condition. During recovery after compression, the pressure difference between the inside and outside of HHSs actively drove the cell suspension into their $V_1$ and $V_2$, similar with diastole of heart. Subsequently, hollow channels of $V_1$ entrapped the cells so that they would not be removed by rinsing[7]; while cells in $V_2$ could be easily removed by rinsing. Besides, the compression stains and cycles of compression-recovery controlled the pressure difference between the inside and outside of HHSs, and thus further regulating the number of loaded cells. Thus, $V_1$ of HHSs responded to mechanical stimulation with high tailorability, realizing a rapid, uniform, and precise manner to load cells without any negative effects. In general, HHSs needed to be immersed in a medium to reach the swelling equilibrium under a static condition before cell loading, causing $V_2$ full of medium. When cells were directly seeded on the HHSs, it would be relatively difficult for them to enter $V_2$, causing insufficient cell deposition and ununiform distribution in HHSs after a period of culture, which was consistent with the results from a previous report[43]. By $V_2$-mechanical response, medium could be drained out from $V_2$, differential pressure inside and outside drove the uniform filling of the $V_2$ with cell suspension, subsequently the cells could adhere in the frame of $V_2$ after a period of culture, contributing to a uniform distribution and high loading efficiency of cells in HHSs. Thus, on basis of the post-bioprinting strategy, combination of utilization of $V_1 + V_2$-mechanical response can realize the precisely partitioned loading of multiple cell types, bypassing the harsh requirements for bioinks in extrusion bioprinting. Notably, the post-bioprinting strategy for cell loading can be extended to larger sizes of HHSs by increasing compression strain and cycles.

Compared with the HHSs, better osteogenesis was achieved in HHSs-cells in evaluation of results in vivo, demonstrating the effectiveness of HHSs-cells for treatment of challenging bone defects. On one hand, HHSs-cells could provide abundant crucial cells for osteogenesis, thus promoting the new bone formation[6,44,45], while did not exhibit excessive inflammation/foreign body response. Notably, it was verified that vascularization could accelerate bone formation in challenging bone defects[8,46,47]. BMSCs could not only differentiate into osteoblasts, but also upregulate the vascularization[48]. In segmental bone defects, HHS-M and HHS-ME could significantly facilitate blood vessels formation in an early stage. Moreover, HHS-ME seemed to perform better than HHS-M in this study, which was in line with previous findings that co-culturing of BMSCs and ECs facilitated osteogenesis and angiogenesis compared to BMSCs only[7,48,49]. Whereas, in contrast to other cell-free constructs, such as 3D printed scaffolds, porous scaffolds, and hydrogels, HHS-cells also exhibit a comparable repair effect in rats segmental bone defects[3,50–52]. In addition, compared to the senescent BMSCs in osteoporotic bone, healthy BMSCs not only supported differentiation into osteoblasts, but also inhibited osteoclast activation, thus rebalancing bone formation and resorption in osteoporotic bone defects[53]. On the other hand, compared to the blank group, the cell-free HHSs exhibited an improved ability for bone regeneration, suggesting pure HHSs also provided favorable microenvironments for enhanced osteogenesis. It has long been a consensus that the cell fate commitment of BMSCs is uncertain after transplantation[53]. In order to maximize the function of BMSCs, material design is also necessary in bone tissue engineering[1]. Bioactive ingredients have been commonly integrated into materials to enhance osteogenesis and angiogenesis[21,22,50,54]. GLN-based HHSs containing the bioactive Mg and Si could facilitate osteogenesis and angiogenesis[15,55,56]. Moreover, the hollow structures of HHSs could also promote mass transfer of nutrient and oxygen, as well as the ingrowth of blood vessels to accelerate bone formation[57]. Together, in the study, this post-bioprinting strategy combines the benefits of loaded cells and bioactive substances to reconstruct challenging bone defects successfully.

The proposed post-bioprinting strategy has the potential to be developed as a universal approach for tissue-engineered therapy repairing other tissues, such as cartilage, muscle, nerve, etc. Moreover, it can be also developed to deliver bioactive factors or drugs for cell-free therapy. Nonetheless, it is still in its infancy and lots of challenges should be further investigated to widen the range of its applications. First, the biological mechanism of bone repair by HHS-cells in the study is still unclear, and in-depth and longer-term studies are needed. Second, this post-bioprinting strategy is limited to a certain extent due to the choice of biomaterial inks, which remains questionable whether it is suitable for other tissues. In addition, only two types of cells were partitioned in HHSs in this work, and precise loading of more cells type by this post-bioprinting strategy should be further developed.

In conclusion, a series of large-sized and sophisticated HHSs with tunable hollow structures were successfully fabricated through a simple approach of one-step coaxial printing without supporting materials. The obtained HHSs exhibited excellent mechanical outcomes: attractive resilience, rapid shape recovery, and exceptional fatigue resistance, revealing their drastic mechanical responsiveness. This unique responsiveness was further utilized to realize cell loading into HHSs in a rapid, uniform, precise and friendly manner. Additionally, cell-laden HHSs could effectively repair the critical-sized segmental and osteoporotic bone defects in vivo. Therefore, this mechanical-assisted post-bioprinting strategy offers a universal, efficient, and promising approach to perform cell-based regenerative therapies.

## Methods

### Preparation and characterization of gelatin methacryloyl/Laponite nanoclay/N-acryloyl glycinamide hybrid inks and hydrogels

A series of GLN hybrid inks with variable mass ratios of gelatin methacryloyl/Laponite nanoclay/N-acryloyl glycinamide were prepared. Gelatin methacryloyl was synthesized as described below. In brief, type A porcine skin gelatin (V900863, Sigma-Aldrich, USA) was dissolved in phosphate buffered saline at 50 °C. Subsequently, methacrylic anhydride was slowly added to the gelatin solution and reacted at 50 °C for 3 h. The solutions were dialyzed against distilled water by using 8–14 kDa cutoff (Solarbio, China) dialysis tubing at 40 °C for 3 days and lyophilized for 4 days to obtain gelatin methacryloyl. Then, gelatin methacryloyl, Laponite nanoclay (BYK, Wesel), and N-acryloyl glycinamide (BD1188660, Bidepharm, China) were dissolved in deionized water and adequately mixed at final concentrations (w/v) of 12% gelatin methacryloyl, 10% Laponite nanoclay and 0%, 4%, 8%, 12% N-acryloyl glycinamide to prepare the GLN0, GLN4, GLN8, and GLN12 hybrid inks, respectively. Subsequently, the photoinitiator, phenyl-2,4,6-trimethylbenzoyl phosphate lithium (TCI, L0290) was added to a final concentration of 0.1% (w/v). Finally, the GLN hydrogels were obtained by exposure of GLN hybrid inks in a UV crosslinker for 40 min (Spectronics Corporation, USA). gelatin methacryloyl and N-acryloyl glycinamide were characterized by NMR (Supplementary Fig. 1).

### Rheological test of GLN hybrid inks

The rheological properties of GLN hybrid inks were conducted using a shear rheometer (MCR 302, Anton Paar, Austria). To assess shear thinning properties, viscosity was measured as a function of shear rate $0.1–10 \text{ s}^{-1}$ at 25 °C. Shear recovery was assessed by applying shear rate sweeps at three stages with shear rate $0.1 \text{ s}^{-1}$ (60 s) −1 or 10 or $50 \text{ s}^{-1}$ (5 s) − $0.1 \text{ s}^{-1}$ (60 s), $T = 25$ °C. Strain-yielding behavior was assessed using a strain sweeps test, and the shear strain was changed successively from strain 1% (90 s) to strain 50%, 100%, 150% (90 s), with f = 1 Hz and $T = 25$ °C.

## Morphology observation of GLN hydrogels

The GLN hydrogels were sharply frozen in liquid nitrogen and lyophilized immediately. The cross sections of the freeze-dried samples ($n = 3$) were observed under a field emission scanning electron microscopy (SEM, Hitachi S-4800, Japan). In addition, the elemental analysis (Si and Mg) of the GLN hydrogels were detected using an energy dispersive spectrophotometer (EDS, Hitachi, Japan).

## Mechanical properties of GLN hydrogels

All the GLN hydrogels were immersed in PBS to reach swelling equilibrium before the test. The compression modulus and strength of GLN hydrogels (φ 8 mm × h 10 mm, $n = 4$) were determined using a mechanical analyzer (Care, IBTC-300SL, China) at room temperature and the crosshead speed at 0.05 mm/s. The modulus of GLN hydrogels was obtained by the initial (straight line) linear slope of the stress-strain curve. The compressive strength was the peak of stress from the stress-strain curve. The stress relaxation was measured under 20% strain, and the relaxation time was defined as the time that the stress reduces to half of the initial stress.

## Water absorption of GLN hydrogels

The water adsorption of GLN hydrogels was measured by incubating the hydrogels (φ 10 mm × h 0.5 mm, $n = 4$) in PBS at 37 °C for 12 h to reach swelling equilibrium. Wet weight ($W$) was recorded at preset time points. The rate of water adsorption was defined as $(W-W_O)/W_O \times 100\%$, where $W_O$ was the initial wet weight of the samples. Besides, the gross morphology was imaged with a digital camera.

## Degradation of GLN hydrogels

The degradation of GLN hydrogels was measured by incubating the hydrogels (φ 10 mm × h 0.5 mm, $n = 4$) in PBS at 37 °C. Wet weight ($W$) was recorded at preset time points. The degradation rate of was defined as $(W_O-W)/W_O \times 100\%$, where $W_O$ was the initial wet weight of the samples.

## Water contact angles test of GLN hydrogels

The water contact angle of the GLN hydrogels was examined with an automatic video micro contact angle-measuring instrument. The deionized water falling on top surfaces of samples ($n = 4$) had a volume of 5 μL and a velocity of 1 μL s$^{-1}$, then images of droplets were recorded by microscope lens and a camera. Analysis and processing software was used to calculate the contact angle of the GLN hydrogels.

## Cell viability, spreading, proliferation, and osteogenic differentiation of cells on GLN hydrogels

Human bone marrow mesenchymal stem cells (hBMSCs) (Cyagen, HUXMF-01001, Guangzhou) were cultured in a-MEM containing 10% fetal bovine serum (FBS, Gibco) and 100 U/mL penicillin/streptomycin in a humidified atmosphere with 5% CO$_2$ at 37 °C. hBMSCs at passage 6 (P6) were used for further experiments. The hBMSCs were seeded onto the sterilized GLN hydrogels (φ 10 mm × h 0.5 mm) placed in 24-well cell culture plates (Corning) at a seeding density of $2 \times 10^4$ cells/well. Live/dead viability was used for determining cell viability. The samples cultured for 1 day were stained with 5 μg/mL fluorescein diacetate (FDA, Sigma) and 5 μg/ml propidium iodide (PI, Sigma), after washing the samples with PBS, cell viability in the hydrogels was observed visually with a confocal laser scanning microscopy (CLSM, Leica SP8, Germany). The samples cultured for 1 day were fixed in 4% paraformaldehyde for 10 min at 4 °C. After cell permeabilization and blocking with 0.1% Triton-X and 10% goat serum, respectively. Then, the samples were incubated with an anti-vinculin antibody (Abcam, ab129002) at 1:100 dilution in 1% BSA-containing PBS solution at 4 °C overnight. Subsequently, the samples were washed 5 times by flowing PBS and treated with the goat anti-rabbit IgG Alexa Fluor 488

conjugate secondary antibody (Abcam, ab150077) at 1:500 dilution for 1 h at room temperature. Meanwhile, to visualize cell spreading, the samples were incubated in 5 U/mL Alexa Fluor-594 phalloidin (Invitrogen, A12381) for 45 min at room temperature, and cell nuclei were stained with Hoechst 33342. Finally, the stained samples were examined under a CLSM. For quantitative analysis of cell proliferation, the samples were incubated in a serum-free medium containing 10% CCK8 (Dojindo, Japan) and measured using a multi-detection microplate reader (Bio-Rad 550). Cell proliferation was analyzed on 1, 3, and 5 days. The activity of alkaline phosphatase was detected with a BCIP/NBT staining kit (Beyotime, C3206) according to the manufacturer's protocol. Briefly, the samples cultured for 7 days were fixed in 4% paraformaldehyde for 10 min at 4 °C and incubated with BCIP/NBT working solution at 4 °C overnight. Then, the samples were washed three times with deionized water. The images were acquired by an optical microscope. The samples cultured for 14 days were harvested for detecting collagen I (Col I). Immunofluorescence staining was performed as described above. The primary antibody and secondary antibody were anti-Col I at dilution 1:200 (Abcam, ab34710) and goat anti-rabbit IgG Alexa Fluor 488 at 1:500 dilution (Abcam, ab150077), respectively. Osteogenic gene expression (ALP and COLI) was assessed by qRT-PCR. At 7 days, the samples ($n = 3$) were collected. Total RNA was extracted according to the manufacturer's instructions for the RNeasy Mini Kit (Qiagen, 74904). And then, the extracted RNA was reverse-transcribed into cDNA using the RevertAid First Strand cDNA Synthesis Kit (Invitrogen, K1622). Quantitative real-time PCR was performed using the CFX96TM real-time PCR detection system (Bio-Rad, USA) with RealStar Fast SYBR qPCR Mix (Genstar, A301). The cycling conditions for qRT-PCR are 95 °C for 2 min, followed by 40 cycles of 95 °C for 15 s, 60 °C for 15 s, and 72 °C for 30 s. And the gene expression level of each targeted gene was determined by the $\triangle\triangle$Ct method, and GAPDH was used as a housekeeping gene to normalize the results. The sequences of the primers are given in Supplementary Table 2.

## Fabrication of HHSs

The HHSs were one-step printed through a coaxial needle using a 3D printer (BioScaffolder 3.1, GeSiM, Germany) at room temperature. The prepared GLN ink was loaded into a printhead and connected to the external nozzle of a coaxial needle, while the internal nozzle was connected to an empty printhead. During the printing process, the GLN ink was directly extruded from the coaxial nozzle to form hollow filaments with an air pressure of 300-500 KPa, and then constructs were fabricated at a printing speed of 6 mm/s. Hollow filaments with tunable inner and outer diameters were controlled by the size of coaxial needles. To further fabricate the constructs with complicated and irregular shapes, bone-shape constructs were generated by a 3D modeling software (3D Studio Max). Subsequently, the 3D models were uploaded onto GeSiM Robotics Software, which generated 2D and 3D views of structures and G-codes to print. A digital camera (Nikon, Japan) was used to acquire images of the printed constructs. Finally, the HHSs were exposed in a UV crosslinker for 80 min with glycerin moisturizing.

## Characterization of HHSs

The morphologies of filaments and grids of HHSs were observed by a stereo-zoom microscope (Hirox RH-2000, USA). Hollow structures were displayed by immersing the HHSs in water, and the images were captured using a digital camera. For further characterization of the internal structures of HHSs, samples were sliced to ~200–300 μm in thickness and labeled with Rhodamine B. After staining for 1 min, slices were triply washed by deionized water and imaged with an inverted fluorescence microscope (Nikon, Japan). The average size of inner and outer diameters of one filament, and distances between two filaments in a layer were defined as $d$, $D$, and $L$ respectively. Quantification was performed in ImageJ.

## Mechanical properties of HHSs

The HHSs ($10 \times 10 \times 10$ mm) were immersed in PBS for 1 h to reach a swelling equilibrium before tests. Compression resilience of HHSs with various inner diameters ($L_{0.4}D_{0.6}d_z$, $z = 0$, 0.2, 0.3, 0.4, $n = 3$) were determined using a mechanical testing instrument (Care, China) with a crosshead speed at 0.1 mm/s at room temperature. A strain of 80% was set for the compression-recovery tests, and the whole process was recorded by a digital camera. The compression modulus of HHSs was determined from the linear slope of the stress-strain curve at a strain of 5%. For characterization of resilient speed of HHS, luminous fluid was added onto the HHSs ($L_{0.4}D_{0.6}d_{0.4}$) to visualize the compression-recovery process, which was recorded by a digital camera.

Fatigue resistance of HHSs was assessed by a cyclic compression test. The HHSs with tunable inner diameters $L_{0.4}D_{0.6}d_z$ ($z = 0$, 0.2, 0.3, 0.4, $n = 3$) were subject to $10^4$ cycles at a frequency of 0.05 Hz with 40% compression strain. The HHSs were fixed in a water bath at room temperature during the test to avoid the influence of moisture evaporation from hydrogels. The test would be aborted if the HHSs were obviously fractured. The morphologies of HHSs before and after testing were respectively captured by a digital camera and stereo zoom microscope.

## Tests of mechanical responsiveness

The mechanical responsiveness of HHSs ($10 \times 10 \times 10$ mm) to a solution was assessed as below. HHSs with tunable inner diameters ($L_{0.4}D_{0.6}d_z$, $z = 0$, 0.2, 0.3, 0.4) were used and distances between two hollow filaments in same layer were also varied ($LxD_{0.6}d_{0.4}$, $x = 0.2$, 0.4, 0.6). Firstly, the HHSs were immersed in water at 37 °C for 8 h to reach a stable state under a static condition. Wet weight (W) was recorded at preset time points. The water uptake ratio was defined as $(W\text{-}W_0)/\rho/(l \times w \times h) \times 100\%$, where $W_0$ was the initial wet weight of the samples, $\rho$ was the value of water density, and $l$, $w$, and $h$ were length, width, and height of the HHS respectively. Subsequently, HHSs were handled with mechanical stimulation in a dynamic condition, and compression with different displacements (20%, 40%, 60%, and 80%) were performed on the HHSs. The wet weight ($W'$) was recorded after the shape recovery of HHSs to the initial state. In addition, the HHS ($L_{0.4}D_{0.6}d_{0.4}$) was further subject to incrementally numbers of cycles of compression-recovery (1, 3, 5, and 10) with 20% and 40% compression strains. The water uptake ratio was defined as $(W'\text{-}W)/\rho/(l \times w \times h) \times 100\%$. All experiments were performed in triplicate to obtain the average data.

To further visualize the mechanical responsiveness of HHSs, HHSs with tunable inner diameter ($L_{0.4}D_{0.6}d_z$, $z = 0$, 0.2, 0.4, 0.6) were chosen. After immersing HHSs in a solution containing 0.5 mg/mL fluorescent microspheres (Aladdin, M122063, China) for 5 min, HHSs were subsequently transferred to an IVIS imaging system (Caliper Life Sciences, USA), and the images before or after immersing were acquired at Ex = 605 nm, Em = 680 nm. Next, HHSs were transferred back to the solution and subject to incremental compression displacements (20%, 40%, 60%, and 80%) and number of cycles of compression-recovery (1, 3, 5, and 10) with 20% and 40% compression displacement. The images of HHSs were also captured using an IVIS imaging systems after each process of compression-recovery. All experiments were performed in triplicate to obtain the average data.

## Cell loading to HHSs

**Cell culture.** *Human* bone marrow mesenchymal stem cells (hBMSCs, HUXMF-01001, Cyagen), hBMSCs-GFP (HUXMA-01101, Cyagen), and *rat* bone marrow mesenchymal stem cells (rBMSCs, were cultured in α-MEM (Hyclone)) containing 10% fetal bovine serum (FBS, Gibco) and 100 U/mL penicillin/streptomycin. Human umbilical vein endothelial cells (HUVECs, e005, iCell), HUVECs-RFP (h111, iCell) and *rat* umbilical vein endothelial cells (rECs, e004, iCell) were cultured in a special culture medium supplied from the iCell. All cells were cultured in a humidified atmosphere with 5% $CO_2$ at 37 °C. rBMSCs were extracted

from long bones of limbs of *rats*. Briefly, alpha-modified Eagle's medium (α-MEM, HyClone) containing 20% fetal bovine serum (FBS, Gibco) and 100 U/mL penicillin/streptomycin (Hyclone) was injected using syringes to flush cells out of marrow cavities. After 24 h, the medium was replaced with fresh culture medium to remove nonadherent cells. The cells were sub-cultured in α-MEM containing 10% FBS and 100 U/mL penicillin/streptomycin. The sterilized HHSs ($L_{0.4}D_{0.6}d_{0.4}$, length 10 mm, width 10 mm, height 10 mm) were immersed in α-MEM medium for 1 h at 37 °C before cell loading.

**Rapid Loading.** Cell loading into HHSs by mechanical assistance was investigated. Firstly, two approaches were used: The first one was direct immersion of HHSs in hBMSCs cell suspension ($1 \times 10^6$ cells/mL) for 4 s under a static condition; while the second one utilized a compression-recovery cycle in hBMSCs cell suspension under 80% strain within 4 s ($V_1$-mechanical response). The subsequent HHSs were immediately transferred from the cell suspension and further cultured in medium. After 4 h of culture, the samples were stained with 5 μg/mL fluorescein diacetate (FDA, Sigma), triply washed with PBS, and then visualized by confocal laser scanning microscopy (CLSM, Nikon C2, Japan). The number of cells loaded into HHSs were quantified by ImageJ. Subsequently, the influence of compression displacement (20%, 40%, 60%, and 80%) in hBMSCs cell suspension ($1 \times 10^6$ cells/mL) and cycle times (1, 3, 5, and 10) under 40% compression displacement in hBMSCs cell suspension ($5 \times 10^5$ cells/mL) on cell loading of HHSs were also evaluated respectively. All experiments were performed in triplicate at least.

**Uniform distribution of cell loading.** The cell loading into HHSs were handled in other two distinct means: One was directly seeding of 0.5 mL cell suspension (hBMSCs, $1 \times 10^6$ cells per mL) onto the HHSs; the other was removal of the medium in $V_2$ under 80% compression displacement, and subsequent seeding of 0.5 mL hBMSCs cell suspension onto the HHSs ($V_2$-mechanical response). After 4 h of culture, same characterization of cells was performed as above. Besides, the cells in the surface, middle, and bottom sections of HHSs were also quantified. All experiments were performed with five parallel samples.

**Partitioned loading of multiple cell types.** To achieve partitioned loading of multiple cell types into HHSs, $V_1$ and $V_2$ of HHSs were manipulated by combination of $V_1 + V_2$-mechanical response simultaneously. Firstly, HHSs were subject to a process of compression-recovery under 80% compression displacement in a HUVECs-RFP cell suspension within 4 s. Subsequently, samples were taken out and the grids were triply flushed ($V_2$) with α-MEM medium. After culture for 4 h, the process of $V_2$-mechanical response was similarly performed as described above, except that the hBMSCs-GFP were used instead of hBMSCs. The loading and distribution of HUVECs-RFP and hBMSCs-GFP in HHSs after 24 h of culture were visualized by CLSM.

## Cell viability under mechanical stimulation

Cell damage is typically associated with a decrease in cell mitochondrial membrane potential and an increase in ROS generation besides a reduction in cell proliferation. To assess whether mechanical stimulation have negative influence on cell viability. Mitochondrial membrane potential, ROS and proliferation of the cells were examined. After a process of compression-recovery with 80% compression displacement and 10 cycles with 40% compression displacement in cell suspension, cells loaded in HHSs were collected by compression with 80% compression displacement again. Subsequently, the collected cells were seeded on 24-well cell culture plates at a density of $2 \times 10^4$ cells/well, and cells that did not undergo mechanical stimulation were used as control. The following measurements were performed: Mitochondrial membrane potential of cells cultured for 4 h and 24 h was detected by double fluorescence staining of mitochondria using JC-1

(C2006, Beyotime) according to the manufacturer's instructions, either as green fluorescent J-monomers or as red fluorescent J-aggregates. ROS generated by cells was detected via a fluorescence probe DCFH-DA (35845, Sigma), and cells cultured with 200 μM $H_2O_2$ were used as positive control. After 4 h of culture, the cells were incubated in a 10 μM DCFH-DA solution at 37 °C for 30 min, and triply washed by PBS, and then stained with Hoechst 33342. Images of cells labeled by JC-1 and DCFH-DA were captured using a CLSM, and the intensity of red and green fluorescence was quantified by a gray value analysis in ImageJ. Cell proliferation was determined using a CCK8 assay (Dojindo, Japan) according to the manufacturer's instructions, and analyzed on day 1, 3, and 5 ($n = 3$).

### In vitro functional examination
After 14 days of culture, the samples loaded with HUVECs were harvested to evaluate the maintenance of functions of HUVECs in HHSs for detecting CD31 and VWF immunofluorescence. Samples were triply washed with PBS and then fixed in 4% paraformaldehyde at 4 °C overnight. All samples were dehydrated, embedded in OCT, and sectioned at a thickness of 5 μm. The slices were permeabilized and blocked with 0.1% Triton-X and 10% goat serum, respectively. Then, the slices were incubated with the anti-CD31 antibody (ab28364, Abcam) and the anti-VWF antibody (SC-365712, Santa) at 1:100 dilution in 1% BSA-containing PBS solution at 4 °C overnight. Subsequently, the slices were washed 5 times by PBS and treated with a mixture of goat-anti-rabbit IgG Alexa Fluor 488 (ab150077, Abcam) and goat-anti-mouse IgG Alexa Fluor 647 (ab150115, Abcam) conjugated secondary antibody at 1:400 dilution for 1 h at room temperature, and washed 5 times by PBS. Finally, samples were counterstained with Hoechst 33342 and triply washed with PBS. Fluorescence images were captured via a CLSM.

Samples loaded with hBMSCs were cultured in osteogenic differentiation media. After 21 days of culture, the samples were harvested for alizarin red S (ARS) staining to evaluate the ability of calcification. The samples were handled as described above to acquired slices. Then, the slices were stained with ARS solution at a concentration of 1% (pH = 4.2–4.3) for 5 min, and triply washed by deionized water. The images were acquired by an optical microscope.

### Establishment of challenging bone defect models
To assess the concept of HHS-cells for repair of the challenging bone defects, two typical challenging bone defect models were established: the *rat* femoral segmental and *rat* osteoporotic bone defects. The animal experiments were approved by the Institutional Animal Care and Use Committee of Shenzhen Institute of Advanced Technology, Chinese Academy of Science. HHS-cells samples were cultured for 3 days in vitro before implantation.

### The *rat* femoral segmental defect model
32 male Sprague Dawley (SD) *rats* (12 weeks old) were divided into 4 groups randomly: Blank (no samples), HHS, HHS-M (HHS loaded with rBMSCs), and HHS-ME (HHS loaded with rBMSCs and rECs). A 5 mm critical-sized femoral segmental defect was created according to a previously published procedure[51]. Briefly, anesthesia was performed with 5% isoflurane and $O_2$. Animals were placed in lateral position on a 37 °C warm heating pad, isoflurane was lowered to 2 to 2.5% to maintain anesthesia during the entire surgery. The hindlimb of the *rat* was shaved and disinfected (half of each left and right), and an ~2-cm-long longitudinal skin incision was made, and superficial fascia, biceps femoris and vastus lateralis muscles were separated, exposing the lateral aspect of the femoral bone. After thoroughly cleaning the soft tissue on femur, a seven-hole fixation plate (31 mm in length, 3.5 mm in width, 1.2 mm in thickness, titanium alloy; Baiortho, China) was applied to the lateral aspect of the femur and held in place using forceps and a clamp. At this point, the plate was permanently fixed to the bone using three proximal and three distal

screws (screws, $\varphi = 1.1$ mm; outer screws, 8 mm in length; inner screws, 7 mm in length; Baiortho). Screw holes were created in the femur using drill sleeves and a drill bit ($\varphi = 1$ mm, Baiortho). A 5 mm critical-sized segmental defect was created using a reciprocating saw (Baiortho). Saline was continuously applied during the piercings and osteotomy, and pieces of bone were carefully removed with saline. A defect of ~5 mm was achieved in all animals. All samples used here were designed as cylinders ($\varphi$ 5.5 mm × h 5 mm). The cylindrical samples were push-fitted, and then muscle, fascia and skin were successively closed with sutures. After wound closure, animals were injected with 8 U/mL penicillin and transferred back into their cage. To relieve the pain, all *rats* were orally administrated with ibuprofen (10 mg/kg, PHR1004, Merck) in drinking water for 24 h after surgery.

6 weeks after the defect surgery, randomly selected *rats* (a total of 16, $n = 4$ for each group) were anesthetized. Abdominal cavity was opened, and abdominal aorta and vein were exposed. Then a 1.2 mm angio-catheter was inserted in the aorta and a 1.6 mm angio-catheter was inserted in the vein. 100 mL of heparinized saline (1000 U/ml), 100 ml of 4% paraformaldehyde, and 15 ml of Microfil (Flow Tech, MV-120, USA) were perfused in turn. After being kept for 24 h at 4 °C, the femurs were collected from the rats, and imaged with microcomputed tomography (μCT, SCANCO MEDICAL μCT100, Switzerland) to observe formation of blood vessels.

6 and 12 weeks after the defect surgery, animals were euthanasia using $CO_2$, and the femurs from each group ($n = 4$) were collected and fixed in 4% paraformaldehyde for 48 h. Bone healing in 4 groups were preliminary evaluated by X-ray. Subsequently, fixation plates and screws were removed from the femur before μCT examinations.

### The *rat* osteoporotic bone defect model
A total of 12 female SD *rats* (3 months old) were all subject to ovariectomy surgeries and then raised for another 12 weeks to obtain osteoporotic *rats*. The osteoporotic *rats* were divided into 3 groups at random: Blank, HHS, and HHS-M. All samples used here were designed as cylinders ($\varphi$ 3 mm × h 4 mm). A part of surgical procedures was described above. Two distal femoral metaphyseal critical-size defects ($\varphi$ 3 mm × h 4 mm) were established using a drill in left and right of an osteoporotic *rat*. The eight samples in each group were implanted in the femoral defects. 4 and 8 weeks after of the defect surgery, animals were sacrificed and the distal femurs from each group ($n = 4$) were collected and fixed in 4% paraformaldehyde. BMSCs and OVX-BMSCs were respectively isolated from long bones of limbs of healthy and osteogenic *rats*. Then, alpha-modified Eagle's medium (α-MEM, HyClone) containing 20% fetal bovine serum (FBS, Gibco) and 100 U/mL penicillin/streptomycin (Hyclone) was injected using syringes to flush cells out of marrow cavities. The cells were cultured in a humidified atmosphere with 5% $CO_2$ at 37 °C. After 24 h, the medium was replaced with fresh culture medium to remove nonadherent cells. The cells were sub-cultured in α-MEM containing 10% FBS and 100 U/mL penicillin/streptomycin until passage 2 (P2). The P2 cells were then harvested for further study. BMSCs and OVX-BMSCs were respectively seeded on the 24-well plates at a cell density of $2 \times 10^4$/well. Cell senescence was assessed by the senescence βgalactosidase (SAβgal) staining kit (Beyotime) and qRT-PCR for *p16* gene expression. Cell proliferation was analyzed on 1 and 3 days by CCK8. In addition, the ability of osteogenic differentiation was evaluated by ALP staining on 4 and 7 days and alizarin red (ARS) staining on 14 and 21 days, and qRT-PCR for *Runx2*, *Alp*, *Bmp2*, *Bsp*, *Col 1a1*, and *Ocn* genes expression on 7 days.

### The rat subcutaneous implantation and repair of distal femoral metaphyseal defect
To evaluate inflammation and foreign body response in vivo experiments, three groups: sham, HHSs (length 8 mm, width 8 mm, height 4 mm), and HHSs-M were subcutaneously implanted on the backs of

rats (12 weeks old) under aseptic and anesthetic conditions. Anesthesia was performed with 5% isoflurane and $O_2$. Animals were placed on a 37 °C warm heating pad, isoflurane was lowered to 2–2.5% to maintain anesthesia during the entire surgery. After 7 days of surgery, animals were sacrificed using $CO_2$, and all specimens were collected, fixed in 4% paraformaldehyde, embedded in paraffin, sectioned at 5 μm thickness. The related assessments were conducted including complete blood count, blood biochemistry analysis (liver and kidney function indicators), and histology staining of the heart, liver, spleen, and kidney, as well as immunohistochemical staining of inflammation-related markers IL-6 and TNF-α, and macrophage phenotype related markers CD80 and CD 206. The procedure for immunohistochemical staining was the same as mentioned above. The slices were respectively incubated with an anti-IL6 antibody (bs-6309r) at anti-TNF-α antibody (bs-10802r), anti-CD80 antibody (A001961) and anti-CD206 antibody (AB64693) at 1:200 dilution in 1% BSA-containing PBS solution at 4 °C overnight.

To assess the survival of seeded cells in the HHSs implantation, HHSs-M (length 8 mm, width 8 mm, height 4 mm) with luciferase-overexpressed rBMSCs were subcutaneously implanted in rats. Subsequently, bioluminescence images were captured in rats and a quantitative analysis of bioluminescence was conducted at 1, 4, 7, and 10 days. The images of HHSs were also captured using an IVIS imaging systems.

To assess the hollow structures of HHSs in promoting blood vessel growth and bone formation, two groups of HHSs ($L_{0.4}D_{0.6}d_0$ and $L_{0.4}D_{0.6}d_{0.4}$) were used. Subcutaneous implantation and a distal femoral metaphyseal defect model were established. All animals were 3 months old in the experiments. For subcutaneous implantation, two subcutaneous pockets for implantation of the HHSs (length 8 mm, width 8 mm, height 4 mm) were made on the backs of rats under aseptic and anesthetic conditions. Two groups were respectively transplanted into the left and right ($n = 4$ for each group). After 2 weeks of surgery, animals were sacrificed, and all specimens were collected, fixed in 4% paraformaldehyde, embedded in paraffin, sectioned at 5 μm thickness, and stained with H&E. The surgical procedure for distal femoral metaphyseal defect was the same as the rat osteoporotic bone defect model. A distal femoral metaphyseal critical-size defect (φ 3 mm × h 4 mm) was established in normal rats. Two groups were respectively transplanted into the left and right of the rat's defects at random ($n = 3$ for each group). All samples used here were designed as a cylinder (φ 3 mm × h 4 mm). After 8 weeks of surgery, all animals were sacrificed and the distal femurs for each group were collected, and fixed in 4% paraformaldehyde for 48 h. All samples were scanned using a μCT, then decalcified with EDTA, embedded in paraffin, sectioned at 5 μm thickness, and processed according to standard histological staining procedures. The sections were stained with H&E and Masson's trichrome.

## μCT analysis
All samples harvested from animals were scanned using a μCT with 18 μm resolution and a maximum voltage of 70 KV, to evaluate new bone formation within the defect region. The obtained grayscale images were further reconstructed and analyzed using the Scanco software. During the reconstruction, a global threshold was used to segment the newly formed bone from each implant. The bone volume (BV), bone volume fractions (BV/TV), trabecular number (Tb.N), trabecular thickness (Tb.Th), and trabecular separation (Tb.Sp) were analyzed to evaluate the formation of new bone.

## Histological staining
After the μCT imaging, all samples were decalcified with EDTA, embedded in paraffin, sectioned at 5 μm thickness, and processed according to standard histological staining procedures. The sections were then stained with haemotoxylin and eosin (H&E) to assess general tissue morphology, and Masson's trichrome to observe collagen fibers and visualize bone formation.

## Statistical analysis
All data are expressed as the means ± standard deviation. Statistical analyses were performed using GraphPad Prism 8 software. Statistically significant differences between two experimental groups were assessed using Student's $t$ tests and comparisons among more than two groups were assessed using a Two-way ANOVA. Differences were considered statistically significant when $*p < 0.05$. $**p < 0.01$, $***p < 0.001$, and $****p < 0.0001$ were considered highly significant, ns referred to no significant difference.

## Reporting summary
Further information on research design is available in the Nature Portfolio Reporting Summary linked to this article.

## Data availability
All the data supporting the results in this study are available within the paper and its Supplementary Information files. Any additional requests for information can be directed to, and will be fulfilled by, the corresponding authors. Source data are provided with this paper.

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

## Acknowledgements

The authors gratefully acknowledge the support for this work from the National Key R&D Program [Grant No. 2022YFB3804403 and 2018YFA0703100]; the National Natural Science Foundation of China [Grant Nos. 32122046, 82072082, 32000959, and 32201097]; the Shenzhen Medical Research Fund [Grant Nos. A2303016]; and the Shenzhen Fundamental Research Foundation [Grant Nos. JCYJ20200109114006014, JCYJ20210324115814040 and JCYJ20210324113001005].

## Author contributions

J.Y. and C.R. conceived and designed the study. J.Y, Z.C, X.P, X.W. and H.S. performed in vivo experiments. J.Y. performed the chemical synthesis. J.Y., Z.C., C.G. and J.L. performed in vitro characterization and cell experiments. J.Y. and C.R. analyzed the data. J.Y., K.L., G.W., W.L., and C.R. drafted the manuscript. H.P. and C.R. supervised the entire project. All authors discussed the results and commented on the manuscript.

## Competing interests

The authors declare no competing interests.
