## [Peer Review File · Nature Communications]

REVIEWER COMMENTS

Reviewer #1 (Remarks to the Author):

The paper by Ruan and colleagues under consideration at Nature Communications describes an interesting 3D printing platform where hollow scaffolds are formed through coaxial printing using a composite network of GelMA, laponite nanoclays and N-acryloyl-glycinamide. The formulation was studied in great detail to understand the mechanical properties of the formed constructs, with demonstration of high elasticity and toughness after cyclic loading. The authors developed innovative approaches to seed multiple cells types into the structures and demonstrating some striking behavior with respect to proliferation and differentiation with cyclic loading. The resultant materials were tested in a critical sized bone defect model with demonstrated improvements in tissue regrowth/integration. This is a strong paper with potentially exciting and useful results. However, there are several areas of confusion and some additional work that is necessary before further consideration.

- 1) The authors should consider reducing the number of acronyms in the article as it was difficult to follow and I found myself often looking back in the text repeatedly to remind what they were for. HHS is fine, but some things like chemical names do not need to be acronyms.
- 2) It was difficult to understand the materials chemistry without a clear depiction of chemical structures. The authors should add a panel to figure 1 to highlight what the polymers are with a simple representation of the resultant composite structure.
- 3) One of the most striking attributes of this work is the mechanical performance and shape fidelity of the materials. However, the discussion of molecular interactions is speculative as written with some references to earlier works. The authors should discuss precise molecular mechanisms, pointing to relevant literature to explain the materials behavior. The depictions in figure 3e are intended to demonstrate some sacrificial bonds responding to force. This is not clear in the schematic and they should revise for clarity with clear specific molecular depictions.
- 4) At the beginning of the cell culture work the type of cells used is not defined in the text. Please define at the beginning. Furthermore, the experiments here are quite interesting but very difficult to follow as written. I spent a lot of time going between figures text and supplementary videos but I still do not have a clear picture. I recommend this section be rewritten and simplified for clarity.
- 5) Related to this, it is not clear how the proliferation rate compares between the static and strained conditions. How does the proliferation compare at day 4 for static versus strained? Furthermore, I found this result quite perplexing. Shear stress has been shown to influence proliferation but the large increase on account of seconds of strain is quite surprising and needs to be discussed further.
- 6) There is little discussion about the cells adhere to the material. Considering the components it is not surprising that cells adhere, but if the cells are simply immersed in cell suspensions I would expect wide variability in where they adhere. I believe the latter half of Figure 5 is demonstrating a procedure to rectify this and foster more uniform adhesion, but as written this does not come across clearly and needs more explanation. It is also not clear here how compression helps the process. Is it because the V2 cells are removed, compression then triggers proliferation of V1 cells, then new cells are added to ensure uniform coating? This needs to be explained better.
- 7) When the second cell type is added, how do the authors ensure that they only reside in one area versus another? The image in 5f is clear but the spatial localization of these different cells should be

quantified in a way to demonstrate uniformity and repeatability.

8) The cell viability results in supplementary figure 7 are not very clear. It would be better to present this data as %viability normalized to control using live/dead or similar. It took me too long to puzzle out what the red-green ration was indicating. In addition the ROS panel is very confusing and it is not clear what is being shown. More detail on these experiments is necessary.

9) The differentiation results are interesting but it is difficult to understand how the experiment was performed to trigger osteogenesis and also how the samples were prepared for analysis. While some of this is in the methods it should be briefly described when first presented in the results.

10) The defect model and osteoporotic experiment is compelling. However, I was looking to see how the hollow channels might enhance cell ingrowth and new tissue formation. Was this observed? The histology is clear but it would be useful to see a closer view of new cell infiltration across the different conditions. Related to this I am wondering if the seeded cells survive implantation and whether they engraft with the native tissue.

11) The authors should discuss whether any overt inflammation/foreign body response was observed across their in vivo experiments.

Reviewer #2 (Remarks to the Author):

In this paper, the GelMA/NAGA/Nanoclay hollow hydrogel scaffold was assembled into a cube scaffold by 3D printing, and the cell suspension was adsorbed by the excellent elastic deformation ability of the scaffold, so as to reduce the shear damage to the cells caused by the direct loading cell printing process. At the same time, the two spaces of the cube scaffold (the internal space of the grid and the hollow pipe) are used to realize the regional load of the two cells.

The research has some new ideas, however, from the perspective of materials, this material has been comprehensively analyzed in the author 's previous article. The first half of this article does not have more prominent highlights. From the application point of view, the author used the cell-loaded scaffold to carry out two kinds of bone defect model repair experiments. Obviously, it highlights the advantages of carrying two kinds of cells, but it does not have a closer relationship with the physical properties of the material itself, and does not show a stunning effect and advantages different from the existing research. Furthermore, the way of loading cells in this article does not seem to be the first in this field. By squeezing out the water of the material itself to adsorb cells, such a way does not seem to be called ' mechanical response '.

1. By squeezing and rebounding, the material is loaded with two kinds of cells in different spaces. This process, I think, is not clear in the Figure, text and video. In addition, whether the process of loading BMSCs into the V2 space has an effect on the loaded HUVECs.

2. If the cell-material scaffold has advantages, what is the evidence that the implanted cells play a role in the defect, and what is the fate of the cells loaded in vivo. The authors did not give more detailed data in animal test, only routine characterization.

3. How to define mechanical response? I think it is just a new way of cell loading, similar to sponges. In addition, whether this loading method has advantages in cell loading rate and activity compared with other existing cell loading methods.

4. Whether the adaptation of mechanical properties of materials under different bone defect models should be considered. Moreover, the degradation and swelling of the materials need to be supplemented.
5. The author analyzes the LDX of the printed structure, but the impact of these parameters is very confusing, and it is not clear why the author finally chose one set of parameters.
6. The author has published relevant articles on this material, and the physical and chemical properties of the material have been fully analyzed and tested. Why the first half of this article focuses on optimizing the parameters.
7. The author finally chose GLN12, only from the structure to determine whether it is too single. Whether the biological effects of different concentrations of materials are different.
8. What are the advantages of this bone defect repair materials compared with the existing research.
9. The abstract and the introduction do not reflect the highlights of the article, the advantages of the material.
10. How long the seed cells were implanted in the defect site after in vitro inoculation, whether the author needs to give a judgment and whether it affects the repair process.

Reviewer #3 (Remarks to the Author):

The manuscript entitled „A mechanical-assisted post-bioprinting strategy for challenging bone defect repair” by Jirong Yang et al. is of interest for the readers of Nature Communications.

The authors developed a coaxial bioprinting technique that allowed the printing of scaffolds that incorporate thin hollow tubes in high resolution. Different variations of a basic architecture were printed and extensively analysed. Furthermore, different mixtures of the bioink consisting of GelMA, nanoclay and NAGA were also evaluated. The experiments revealed an optimal architecture and bioink composition. Cytocompatibility was assessed using HUVEC and human bMSC. Generally, the high compressability of the scaffold followed by restorage of its initial form allowed effective cell loading. Moreover, cells survived, kept their differentiation and remained proliferative as proved by various standard experiments.

The experiments were completed by two animal studies in order to analyse the value of the scaffold (with or without MSC, EC, MSC+EC) for treatment of large bone defects, respectively for treatment of osteoporotic lesions thereby using rat models. Scaffold cylinders, either empty as control or loaded with cells were placed into the bone defect and vascularization and bone healing was assessed. It could be shown that vascularization was improved if EC were present, the vascular network reflects thereby impressively the architecture of the scaffold, demonstrating the beneficial effect of the transplanted EC on the formation of the new vasculature. Bone healing was assessed 12 weeks after operation. Constructs loaded with MSC, respectively MSC and EC lead to a good bone healing (increasing bone volume, trabecel density) whereas the cell free scaffold did not. Unfortunately biomechanical tests were not performed. Similar bone healing results were seen in the osteoporosis model.

All in all an extensive and comprehensive manuscript is provided. The data are original as proved by pubmed and google scholar research. No signs of data manipulation were obvious. The integration of small hollow tubes provides new, unique characteristics to the ‘optically’ conventional scaffold design which probably ease processes such as cell loading or fitting into bone defects. Despite these

advantages, the regenerative potential seems in my eyes comparable to similarly designed 3D printed scaffolds.

Although a very well elaborated study is presented, some questions/suggestions remain as pointed out in the following.

1. It is only a formal aspect but a page numbering would facilitate the review of the manuscript.
2. It can be assumed that the improved compressibility is a consequence of the hollow tubes. This should be more clearly stated in the manuscript.
3. Cell loading by repeated compression and suction: With the scaffold volume of 1 cubic centimeter, one compression cycle seems to be sufficient to draw enough cells also into the interior of the scaffold. Are multiple cycles necessary for larger volumes? And: How quickly are the cells adherent, are they not flushed out of the material again by another compression cycle? Have you performed any analyses on this?
4. Are cells also settled in the tube structures during cell loading by compression and subsequent suction of the cells? Based on the photographs (Fig. 5), this could be assumed. Is cell survival and proliferation altered within the tubes versus cells on the structures?
5. Mechanical properties of HHSs: Are the numbers provided for "z" in the unit mm?
6. Cell viability under mechanical stimulation: What is the cell loss during compression cycles? Or does 40% compression did not lead to cell flushing out of the scaffold?
7. P6, l16: avoid ratings such as "attractive" in the results section.
8. P9, l23 (Materials and Methods section) source or strain of hBMSC-GFP?
9. Discussion p15, l4-6: Scalability, is cell loading generally also applicable in larger scaffolds? How compression strain and cycles have to be adjust?
10. Figure 3b: Please remove the heart images. It is just an analogy and does not provide relevant new information.
11. Supplemental images: It is stated in results section (p5l12) that "...formed tailed bridges between nanoclay structures". Bridges are obviously seen but also in the control without nanoclay. To be honest I am not sure if I can see nanoclay structures. Please provide additional images in higher magnification or mark the structures that are supposed to be nanoclay structures.
12. Discussion: Please compare bone healing response with other, more 'conventionally' 3D printed scaffolds. In my opinion bone healing results are comparable to other solutions especially when considering the long period of 12 weeks.

Detailed response to Reviewer comments:

Dear Reviewers:

We are very grateful for the support and suggestions on our manuscript provided by the reviewers. Our manuscript has been carefully revised according to these valuable comments. Major changes have been highlighted in **yellow**. The point-by-point response to the comments is as follows:

Reviewer #1 (Remarks to the Author):

The paper by Ruan and colleagues under consideration at Nature Communications describes an interesting 3D printing platform where hollow scaffolds are formed through coaxial printing using a composite network of GelMA, laponite nanoclays and N-acryloyl-glycinamide. The formulation was studied in great detail to understand the mechanical properties of the formed constructs, with demonstration of high elasticity and toughness after cyclic loading. The authors developed innovative approaches to seed multiple cells types into the structures and demonstrating some striking behavior with respect to proliferation and differentiation with cyclic loading. The resultant materials were tested in a critical sized bone defect model with demonstrated improvements in tissue regrowth/integration. This is a strong paper with potentially exciting and useful results. However, there are several areas of confusion and some additional work that is necessary before further consideration.

Generally Respond: Thank you very much for the reviewer's recognition of our work. According to the comments of the reviewer, we have carefully revised the manuscript and added some additional data. The point-by-point response to the reviewer's comments is as follows:

1) The authors should consider reducing the number of acronyms in the article as it was difficult to follow and I found myself often looking back in the text repeatedly to remind what they were for. HHS is fine, but some things like chemical names do not

need to be acronyms.

Respond: According to the reviewer's suggestions, we have reduced the number of acronyms in the revised manuscript to make it easy to follow. Some acronyms of chemical names such as GelMA and NAGA have been replaced by the full name.

2) It was difficult to understand the materials chemistry without a clear depiction of chemical structures. The authors should add a panel to figure 1 to highlight what the polymers are with a simple representation of the resultant composite structure.

Respond: In order to provide a clear depiction of chemical structures and interactions of GLN ink and hydrogel, we have added Supplementary Fig. 2 in the revised supplementary information. Considering that the image occupies a relatively large space, we have placed it in the supplementary information instead of Fig.1.

Supplementary Fig. 2 The chemical structures and interactions of GLN hybrid ink and hydrogel.

3) One of the most striking attributes of this work is the mechanical performance and shape fidelity of the materials. However, the discussion of molecular interactions is speculative as written with some references to earlier works. The authors should

discuss precise molecular mechanisms, pointing to relevant literature to explain the materials behavior. The depictions in figure 3e are intended to demonstrate some sacrificial bonds responding to force. This is not clear in the schematic and they should revise for clarity with clear specific molecular depictions.

Respond: We have discussed precise molecular mechanisms and pointed to relevant literature to explain the materials behavior in the revised manuscript. And the revised parts as below:

“The sacrificial bonds, such as “house-of-cards” structure of nanoclay³⁶, electrostatic interactions among nanoclay-polymer chains (gelatin methacryloyl or NAGAN-acryloyl glycinamide)^{36,37}, and multiple hydrogen bonding among NAGAN-acryloyl glycinamide molecules³⁸, responsible for energy dissipation, existed in the backbone (GLN hydrogel) of HHSs.”

Moreover, we have revised the Fig. 3e to provide clear specific molecular depictions about sacrificial bonds as below, and detailed chemical structures and interactions of GLN hydrogel in the Fig. 3e can be found in the added Supplementary Fig.2.

Fig. 2e Illustration of mechanism of HHSs for resistance to fatigue. The sacrificial bonds in the backbone of HHSs, including electrostatic interactions and multiple hydrogen bonds, were mostly reversible, and ruptured to dissipate energy during deformation and recovered after relaxation.

4) At the beginning of the cell culture work the type of cells used is not defined in the text. Please define at the beginning. Furthermore, the experiments here are quite

interesting but very difficult to follow as written. I spent a lot of time going between figures text and supplementary videos but I still do not have a clear picture. I recommend this section be rewritten and simplified for clarity.

Respond: We have defined the type of cells used at the beginning, revised the section to simplified for clarity as below. Moreover, we have added Supplementary Video 5.

“As shown in Fig. 5ei, a simple approach of V_2 -mechanical response was further developed to remove medium only in V_2 , subsequently seeding hBMSCs cell suspension, thereby regulating cell number and uniformity. After 4 h of culture, hBMSCs could adhere in the HHS owing to good adhesive properties of GLN hydrogel (Supplementary Fig. 8). It was clearly observed that hBMSCs distributed uniformly in the whole V_2 of HHSs (surface-middle-bottom) (Fig. 5eii,eiii) and the number of hBMSCs loaded in V_2 of HHSs could be improved about 200% (Fig. 5eiv) via the V_2 -mechanical response compared with direct cell seeding, suggesting the availability of V_2 in HHSs. The processes of direct cell seeding and V_2 -mechanical response approaches were displayed in Supplementary Video 4. As shown in Fig. 5f and Supplementary Video 5, by successive processes of loading HUVECs-RFP via V_1 -mechanical response and seeding hBMSCs-GFP V_2 -mechanical response (V_1+V_2 -mechanical response), HUVECs-RFP and hBMSCs-GFP were distributed in V_1 and V_2 , respectively.”

5) Related to this, it is not clear how the proliferation rate compares between the static and strained conditions. How does the proliferation compare at day 4 for static versus strained? Furthermore, I found this result quite perplexing. Shear stress has been show to influence proliferation but the large increase on account of seconds of strain is quite surprising and needs to be discussed further.

Respond: Supplementary Fig. 10d showed cell proliferation after mechanical stimulation with 80% strain and 10 cycles of compression-recovery under 40% strain, the results demonstrated that there was no significantly different proliferation rate of cells undergo strained condition or not. Fig. 5a showed that more loaded cells in the HHS after undergoing strained condition compared to that under static condition at

day 1 rather than reflecting proliferation of cells. Moreover, in order to clarify the proliferation in the HHS under static condition at day 3, we have added the Supplementary Fig. 8 (as below).

Supplementary Fig. 8 Fluorescence image of loaded cells in the HHS under static condition at day 3

6) There is little discussion about the cells adhere to the material. Considering the components it is not surprising that cells adhere, but if the cells are simply immersed in cell suspensions I would expect wide variability in where they adhere. I believe the latter half of Figure 5 is demonstrating a procedure to rectify this and foster more uniform adhesion, but as written this does not come across clearly and needs more explanation. It is also not clear here how compression helps the process. Is it because the V_2 cells are removed, compression then triggers proliferation of V_1 cells, then new cells are added to ensure uniform coating? This needs to be explained better.

Respond: hBMSCs could adhere in the HHS owing to good adhesive properties of GLN hydrogel (Supplementary Fig. 11). Additionally, we have added the explanation in the discussion (P15 L25-26) to clarify how compression helps uniform adhesion

“ By V_2 -mechanical response, medium could be drained out from V_2 , differential pressure inside and outside drove the uniform filling of the V_2 with cell suspension, subsequently the cells could adhere in the frame of V_2 after a period of culture, contributing to a uniform distribution and high loading efficiency of cells in HHSs.”

7) When the second cell type is added, how do the authors ensure that they only reside in one area versus another? The image in 5f is clear but the spatial localization of these different cells should be quantified in a way to demonstrate uniformity and repeatability.

Respond: The detailed process of partitioned loading of multiple cell types is described in the methods part (P21 L23-28) and we have added a video (Supplementary Video 5) to present this process.

“Firstly, HHSs were subject to a process of compression-recovery under 80% compression displacement in a HUVECs-RFP cell suspension within 4 s. Subsequently, samples were taken out and the grids were triply flushed (V_2) with α -MEM medium. Then, the process of V_2 -mechanical response was similarly performed as described above, except that the hBMSCs-GFP were used instead of hBMSCs. The loading and distribution of HUVECs-RFP and hBMSCs-GFP in HHSs after 24 h of culture were visualized by CLSM.”

According to the above description, HUVECs-RFP in the V_2 could be flushed with medium, thus HUVECs-RFP only distributed in the V_1 . Additionally, as shown in Fig. 4a,b, without mechanical stimulation, the water uptake of HHSs sharply raised within the initial 5 min which depended on the grids, and subsequently kept constant over time, suggesting that V_1 didn't adsorb water in the condition. Thus, hBMSCs-GFP only distributed in the V_2 .

Besides, in my opinion, although the partitioned introduction of multiple cell types into scaffolds have been successfully implemented, it is not the key points in this study. We believe that in-depth research to obtain quantitative data on the spatial localization of these different cells should be very important and will be conducted to refine the study in our subsequent work.

8) The cell viability results in supplementary figure 7 are not very clear. It would be better to present this data as %viability normalized to control using live/dead or similar. It took me too long to puzzle out what the red-green ration was indicating. In addition the ROS panel is very confusing and it is not clear what is being shown.

More detail on these experiments is necessary.

Respond: Cell damage is typically associated with a decrease in cell mitochondrial membrane potential and an increase in ROS generation besides a reduction in cell proliferation. We have added this sentence in the Methods (P22 L2-3). In order to make these experiments clearer, we have revised the related parts.

JC-1 is a widely used fluorescent probe for detecting the mitochondrial membrane potential. When the mitochondrial membrane potential is high, JC-1 aggregates within the mitochondrial matrix, forming J-aggregates that emit red fluorescence. Conversely, when the mitochondrial membrane potential is low, JC-1 remains in its monomeric form, emitting green fluorescence as it does not aggregate within the mitochondrial matrix. This convenient fluorescence color shift allows for the detection of changes in mitochondrial membrane potential. Supplementary Fig. 10a shows JC-1 staining, either as green fluorescent J-monomers or as red fluorescent J-aggregates. The quantitative analysis of ratio of red/green reflects that the mitochondrial membrane potential of the cells (Supplementary Fig. 10b) rather than live/dead.

9) The differentiation results are interesting but it is difficult to understand how the experiment was performed to trigger osteogenesis and also how the samples were prepared for analysis. While some of this is in the methods it should be briefly described when first presented in the results.

Respond: We have briefly described in the results. *“Additionally, HHSs loaded with hBMSCs were cultured in osteogenic differentiation media. Together with enhancements in proliferation, osteogenic protein secretion and gene expression of hBMSCs in GLN hydrogels (Supplementary Fig. 11), significant calcium deposition was also observed in HHSs at 21 days of culture (Fig. 5h).”*

10) The defect model and osteoporotic experiment is compelling. However, I was looking to see how the hollow channels might enhance cell ingrowth and new tissue formation. Was this observed? The histology is clear but it would be useful to see a

closer view of new cell infiltration across the different conditions. Related to this I am wondering if the seeded cells survive implantation and whether they engraft with the native tissue.

Respond: As shown in supplementary Fig. 18a and b, hollow channels of HHSs could enhance the ingrowth of native cell and blood vessels, consistent with previous reports (*Biomaterials* 2017, 135, 85-95; *Adv. Sci.* 2017, 4, 1700401). To assess the survival of seeded cells in the HHSs implantation, we subcutaneously implanted HHSs-M with luciferase-overexpressed rBMSCs in rats. Subsequently, we captured bioluminescence images and conducted a quantitative analysis of bioluminescence in rats at various time points (as shown in Supplementary Fig. 13). The results showed that the majority of cells seeded in the HHSs could survive, although a portion of the seeded cells died in vivo.

In addition, the transplanted MSCs, in terms of their local distribution and spatial associations with other types of cells were poorly understood in current studies. Liu et al. (*Stem Cell Reports*. 2022, 17:2318–2333), developed a single-cell 3D spatial correlation (sc3DSC) method to track transplanted MSCs based on deep tissue microscopy of fluorescent nanoparticles (fNPs) and immunofluorescence of key proteins. Locally delivered fNP-labeled MSCs enhanced tibial defect repair, increased the number of stem cells and vascular maturity in mice. fNP-MSCs persisted in the defect throughout repair. The study of cell fate loaded in the scaffold is a long-term and challenging subject, and we will make our best effort to address this key issue in our future research.

Supplementary Fig. 13 The survival of seeded cells in the HHSs implantation. a, Representative bioluminescence images of rats after subcutaneous implantation HHS-M with luciferase over-expressed rBMSCs. **b,** Quantitative analysis of bioluminescence in rats at different time points. Dates are presented as means \pm s.d, n = 5 per group.

11) The authors should discuss whether any overt inflammation/foreign body response was observed across their in vivo experiments.

Respond: To evaluate inflammation and foreign body response in vivo experiments, we conducted assessments including complete blood count, histology staining of the heart, liver, spleen, and kidney, blood biochemistry analysis (liver and kidney function indicators), as well as immunohistochemical staining of inflammation-related markers IL-6 and TNF- α , and macrophage phenotype related markers CD80 and CD206 (as shown in Supplementary Fig. 12). In comparison to HHS without cells, the results indicated that HHS-M did not display excessive inflammation and foreign body response. In addition, we have added related discussion in the P16 L7-8.

Supplementary Fig. 12 The inflammatory and foreign body responses of HHSs-M for rat subcutaneous implantation at 7 days. **a**, H&E staining of the heart, liver, spleen, and kidney. **b**, The complete blood count. **c**, The blood biochemistry analysis, liver function indicators (AST, ALT and TP) and kidney function indicators (CREA and UA). Dates are presented as means \pm s.d, $n = 3$ per group. **c**, Photographs of macroscopic implants, H&E staining, immunohistochemical staining of IL-6, TNF- α , CD206 and CD80.

Reviewer #2 (Remarks to the Author):

In this paper, the GelMA/NAGA/Nanoclay hollow hydrogel scaffold was assembled

into a cube scaffold by 3D printing, and the cell suspension was adsorbed by the excellent elastic deformation ability of the scaffold, so as to reduce the shear damage to the cells caused by the direct loading cell printing process. At the same time, the two spaces of the cube scaffold (the internal space of the grid and the hollow pipe) are used to realize the regional load of the two cells.

The research has some new ideas, however, from the perspective of materials, this material has been comprehensively analyzed in the author 's previous article. The first half of this article does not have more prominent highlights. From the application point of view, the author used the cell-loaded scaffold to carry out two kinds of bone defect model repair experiments. Obviously, it highlights the advantages of carrying two kinds of cells, but it does not have a closer relationship with the physical properties of the material itself, and does not show a stunning effect and advantages different from the existing research. Furthermore, the way of loading cells in this article does not seem to be the first in this field. By squeezing out the water of the material itself to adsorb cells, such a way does not seem to be called ' mechanical response '.

Generally Respond: Thank you very much for the reviewer's comments and suggestions. In our opinions, the key novelty of this work is that a mechanical-assisted post-bioprinting strategy is first presented to realize the cell loading into hydrogel-based scaffolds in a rapid, uniform, precise and friendly manner. This strategy can avoid occurrence of cell damage during the directly bioprinting process and the intrinsically poor mechanical stability of bioprinted cell-laden scaffolds, valuable for broadening the application of bioprinting technology in field of tissue engineering (not limited to bone tissue).

For detailed response to your comments:

Although, the biohybrid hydrogels used in this work is similar with the materials in our published work (Adv. Funct. Mater 2020, 30:2001485), it should not influence the innovation of our work. The reasons are as below:

- (1) First, our main aim in this study is not to develop new materials. Instead, we aim to fabricate large-sized and sophisticated heart-inspired hollow hydrogel-based scaffolds (HHSs) that can possess mechanical responsiveness to realize post-bioprinting. As shown in Fig. 1b, we have printed large-sized and bone-shaped HHSs with high shape fidelity, extending beyond the confines of a simple cubic scaffold. The materials developed by our group can satisfy the requirement to achieve the aim but not unique. We can also replace them by other materials that can meet the demands on the basis of universality of our post-bioprinting. For example, Feinberg et al. also used a common material (collagen) to bioprint components of the human heart (Science 2019, 365:482–487), and Jordan S. Miller et al. also used a common materials system of poly(ethylene glycol) diacrylate and gelatin methacrylate to fabricate multivascular networks and functional intravascular topologies (Science 2019, 364: 458–464)

- (2) Second, in our published work, the materials only for coaxial printing of microtubes for potential regeneration of tubular tissue (without any animal assessments). It is extremely easier than fabrication of large-sized and sophisticated hollow hydrogel-based scaffolds in this study. So far, there is no publication for fabricated large-sized and sophisticated hollow hydrogel-based scaffolds through one-step coaxial printing without any supporting materials.

As a proof of concept, we chose two challenging defects: large-sized segmental and osteoporotic bone defects, which remain a thorny problem in orthopedics. The animal models in this work are challenging and highly significant. The reasons are as below:

- (1) The *in vivo* experiments are in order to verify effectiveness of our post-bioprinting instead of demonstrating the cell-laden hydrogel scaffolds for bone regeneration. Mechanical-assisted post-bioprinting strategy offers a universal, efficient, and promising approach to perform cell-based regenerative

therapies not just bone regeneration. Owing to the limited migration or weak regenerative potential of resident cells in many challenging defects, cell-based therapies is promising approach. As a proof of concept, we chose large-sized segmental and osteoporotic bone defects.

- (2) Cell-laden hydrogel scaffolds by bioprinting for regeneration of large-sized segmental and osteoporotic bone defects in large animal models have not been reported yet, and there are only some cell-laden hydrogels or scaffolds for regeneration of bone defects in small animal models, such as mice, rat and rabbit. For example, Anthony Atala et al. and Yong He et al bioprinted cell-laden constructs for repair of calvarial bone defect of rats (Nat. Biotechnol. 2016, 34:312–319; Nat. Commun. 2022, 13:3597).

Moreover, it is not our purpose to highlight the advantages of carrying two kinds of cells, we highlight a mechanical-assisted post-bioprinting strategy to realize partitioned introduction of multiple cell types into scaffolds. Additionally, we have included a comparative analysis on the advantages of the post-bioprinting compared to other cell loading methods previously reported in terms of time, uniformity, condition, designability and size (as shown in Supplementary Fig. 19). Mechanical responsiveness is the special property of HHS, we utilize this characteristic of the HHS to loading cells, we did not call this process for mechanical response.

Therefore, we still believe that this post-bioprinting strategy can offer a robust way to realize cell-based regenerative therapies and improve their utilization in tissue engineering. What is more, this strategy is also potentially universal, e.g., as a therapy for repairing other tissues, or cell-free therapy to deliver bioactive factors or drugs.

According to the comments of the reviewer, we have carefully revised the manuscript. The point-by-point response to the reviewer's comments is as follows:

1. By squeezing and rebounding, the material is loaded with two kinds of cells in

different spaces. This process, I think, is not clear in the Figure, text and video. In addition, whether the process of loading BMSCs into the V₂ space has an effect on the loaded HUVECs.

Respond: The detailed process of partitioned loading of multiple cell types is described in the methods part “*Firstly, HHSs were subject to a process of compression-recovery under 80% compression displacement in a HUVECs-RFP cell suspension within 4 s. Subsequently, samples were taken out and the grids were triply flushed (V₂) with α -MEM medium. Then, the process of V₂-mechanical response was similarly performed as described above, except that the hBMSCs-GFP were used instead of hBMSCs. The loading and distribution of HUVECs-RFP and hBMSCs-GFP in HHSs after 24 h of culture were visualized by CLSM.*” In order to make it clearer, we have also revised the corresponding results part in the manuscript. “by successive processes of loading HUVECs-RFP via V₁-mechanical response and seeding hBMSCs-GFP through V₂-mechanical response (V₁+V₂-mechanical response), HUVECs-RFP and hBMSCs-GFP were distributed in V₁ and V₂, respectively.” Moreover, we have added a corresponding video to present the process of V₁+V₂-mechanical response ((Supplementary Video 5).

Additionally, we proved that the mechanical stimulation has no negative influences on viability of the cells (Supplementary Fig.10), thus the process of loading BMSCs into the V₂ space has no effect on the loaded HUVECs.

2. If the cell-material scaffold has advantages, what is the evidence that the implanted cells play a role in the defect, and what is the fate of the cells loaded in vivo. The authors did not give more detailed data in animal test, only routine characterization.

Respond: There are many literatures have reported that the MSCs could significantly enhanced healing of bone defects in vivo (eg. *Sci. Adv.* 2020, 6:eaaz6725; *PNAS.* 2010, 107:3305–3310; *Nat. Commun.* 2022, 13:3597.). To assess the survival of seeded cells in the HHSs implantation, we subcutaneously implanted HHSs-M with luciferase-overexpressed rBMSCs in rats. Subsequently, we captured bioluminescence images and conducted a quantitative analysis of bioluminescence in rats at various

time points (as shown in Supplementary Fig. 13). The results showed that the majority of cells seeded in the HHSs could survive, although a portion of the seeded cells died in vivo.

Moreover, the cell fate of transplanted MSCs, in terms of their local distribution and spatial associations with other types of cells were poorly understood in current studies. Liu et al. (*Stem Cell Reports*. 2022, 17:2318–2333), developed a single-cell 3D spatial correlation (sc3DSC) method to track transplanted MSCs based on deep tissue microscopy of fluorescent nanoparticles (fNPs) and immunofluorescence of key proteins. Locally delivered fNP-labeled MSCs enhanced tibial defect repair, increased the number of stem cells and vascular maturity in mice. fNP-MSCs persisted in the defect throughout repair. The study of cell fate loaded in the scaffold is a long-term and challenging subject, and we will make our best effort to address this key issue in our future research.

Supplementary Fig. 13 The survival of seeded cells in the HHSs implantation. **a**, Representative bioluminescence images of rats after subcutaneous implantation HHS-M with luciferase over-expressed rBMSCs. **b**, Quantitative analysis of bioluminescence in rats at different time points. Dates are presented as means \pm s.d, n = 5 per group.

3. How to define mechanical response? I think it is just a new way of cell loading, similar to sponges. In addition, whether this loading method has advantages in cell loading rate and activity compared with other existing cell loading methods.

Respond: In our opinions, mechanical responsiveness is the special property of HHS,

which is utilized by us to loading cells. Compared to other cell loading methods previously reported in terms of time, uniformity, condition, designability and size, we have included a comparative analysis (as shown in Supplementary Fig. 19) on the advantages of the post-bioprinting. Mechanical responsiveness of HHSs can realize rapid, uniform, friendly and precise cell loading for fabrication of large-scale cell-laden constructs.

Supplementary Fig. 19 The comparative analysis on the advantages of the Post-bioprinting among different cell loading methods.

4. Whether the adaptation of mechanical properties of materials under different bone defect models should be considered. Moreover, the degradation and swelling of the materials need to be supplemented.

Respond: We agree with this review that mechanical properties of materials should be considered under different bone defect. However, in this work, we focus on the

mechanical-assisted post-bioprinting strategy that realize the cell loading into hydrogel-based scaffolds in a rapid, uniform, precise and friendly manner rather than materials itself. In order to verify effectiveness and application of our post-bioprinting in tissue regeneration, we chose two challenging defects: large-sized segmental and osteoporotic bone defects, which remain a thorny problem in orthopedics. Moreover, on the basis of the constructive suggestion of the review, we will consider developing other HHS with different mechanical properties to apply in different defect models in the future.

The swelling of the GLN hydrogels is showed in supplementary Fig. 4e, and the degradation of the GLN hydrogels is added in the revised supplementary information. as supplementary Fig. 4e,f. GLN12 barely shrunk and exhibited slow degradation, maintaining their initial shape in PBS solution at 37 °C, which is very important for their corresponding HHSs to keep stable mechanical properties.

5. The author analyzes the LDX of the printed structure, but the impact of these parameters is very confusing, and it is not clear why the author finally chose one set of parameters.

Respond: Fig. 2 shows that tunable hollow structures and grids of HHSs, highlighting that our printed HHSs possessed excellent designability. Fig. 3 reveals that the anti-fatigue performance of HHSs could be enhanced by increasing d , especially the HHSs ($L_{0.4}D_{0.6}d_{0.3}$ and $L_{0.4}D_{0.6}d_{0.4}$) with an exceptional fatigue resistance. Moreover, L and d of HHSs could also remarkably influence the water uptake: the water uptake ratio of L_x HHSs tended to increase and then decrease with L , while the water uptake ratio of d_z HHSs exhibited an increasing tendency with d . Thus, the results reasonably displayed that the water uptake ratio in $L_{0.4}$ HHS was higher than those of $L_{0.2}$ and $L_{0.6}$ HHSs (Fig. 4d). In addition, the water uptake ratio of d_z HHSs exhibited an increasing tendency with d (Fig. 4e). In light of the mechanical properties and water uptake ability, a HHS ($L_{0.4}D_{0.6}d_{0.4}$) was chosen for further assessment.

6. The author has published relevant articles on this material, and the physical and

chemical properties of the material have been fully analyzed and tested. Why the first half of this article focuses on optimizing the parameters.

Respond: In our published work, the materials only for coaxial printing of microtubes for potential regeneration of tubular tissue (without any animal assessments). It is extremely easier than fabrication of large-sized and sophisticated hollow hydrogel-based scaffolds in this study. So far, there is no publication for fabricated large-sized and sophisticated hollow hydrogel-based scaffolds through one-step coaxial printing without any supporting materials. Thus, our primary focus is on optimizing the parameters initially.

7. The author finally chose GLN12, only from the structure to determine whether it is too single. Whether the biological effects of different concentrations of materials are different.

Respond: We chose GLN12 hydrogel based on printability, mechanical outcomes and biological effects.

As described in the results section of “Fabrication of large-sized and sophisticated HHSs”, GLN12 should be the most suitable composition for fabrication of large-sized and sophisticated HHSs and was thus chosen as the basic ink for further assessments in this study. *“To fabricate HHSs, hydrogel inks with excellent printability, appropriate mechanical behaviors as well as biocompatibility were indispensable. N-acryloyl glycinamide possessed multiple hydrogen bonding reinforcements and played a key role in upregulating the printability of GLN inks and mechanical properties of GLN hydrogels (Supplementary Fig. 3). As shown in SEM images (Supplementary Fig. 4), gelatin methacryloyl with relatively high molecular weights formed tailed bridges between nanoclay structures; while N-acryloyl glycinamide with low molecular weights could be physically adsorbed onto nanoclay surface, leading to a decrease of viscosity of GLN inks. Notably, compression modulus and strength of GLN12 could reach to 989 ± 49 KPa and 1200 ± 78 KPa, respectively (Supplementary Fig. 5a,b). Fig. 5e,f displayed that GLN8 and GLN12 barely shrunk and exhibited slow degradation, could maintain their initial shape in PBS solution at*

37 °C, which is very important for their corresponding HHSs to keep stable mechanical properties.

Moreover, as shown in Supplementary Fig. 11, the biological effects exhibited by GLN hydrogels are not entirely the same, while compare to GLN0, GLN4 and GLN8 hydrogels, GLN12 hydrogel could promote proliferation and osteogenic differentiation of hBMSCs.

8. What are the advantages of this bone defect repair materials compared with the existing research.

Respond: We believe that it's necessary to compare the bone defect repair materials with existing research. However, this task poses huge challenges due to variations in animals, models, sizes of bone defects, and standards across different studies. Thus, based on the same animal model (rat segmental bone defects), we have compared bone healing response of HHS-cells in the discussion parts (P16 L13-17) with others. *“Whereas, in contrast to other cell-free constructs, such as 3D printed scaffolds, porous scaffolds and hydrogels, HHS-cells also exhibit a comparable repair effect in rats segmental bone defects^{3, 50-52.}”*

Besides, in this study, we focus on the mechanical-assisted post-bioprinting strategy that realize the cell loading into hydrogel-based scaffolds in a rapid, uniform, precise and friendly manner rather than materials itself. In order to verify effectiveness and application of our post-bioprinting in tissue regeneration, we chose large-sized segmental and osteoporotic bone defects, which remain a thorny problem in orthopedics. We highlight the effect of cell loading on repairing of bone defects. The materials developed by our group can satisfy the requirement to achieve the aim but not unique. We can also replace them by other materials that can meet the demands on the basis of universality of our post-bioprinting.

In addition, we conducted a comparative analysis highlighting the advantages of post-bioprinting over other cell loading methods (Supplementary Fig. 19).

9. The abstract and the introduction do not reflect the highlights of the article, the

advantages of the material.

Respond: As mentioned above, our main focus is not on the materials themselves, so we haven't highlighted it because of the limitation of words number.

10. How long the seed cells were implanted in the defect site after in vitro inoculation, whether the author needs to give a judgment and whether it affects the repair process.

Respond: According to the previous researches (Sci. Adv. 2020, 6:eaaz6725; Biofabrication. 2021, 13: 015011; Nat. Biotechnol. 2016, 34:312 - 319), samples were culture for a period of time in vitro before implantation. Thus, we cultured HHS-cells samples for 3 days in vitro before implantation which mentioned in the Methods part (P23 L18). While, I agree with the review that this process may be affect the repair process, we will do a systematic study in the future work.

+

Reviewer #3 (Remarks to the Author):

The manuscript entitled “A mechanical-assisted post-bioprinting strategy for challenging bone defect repair” by Jirong Yang et al. is of interest for the readers of Nature Communications.

The authors developed a coaxial bioprinting technique that allowed the printing of scaffolds that incorporate thin hollow tubes in high resolution. Different variations of a basic architecture were printed and extensively analysed. Furthermore, different mixtures of the bioink consisting of GelMA, nanoclay and NAGA were also evaluated. The experiments revealed an optimal architecture and bioink composition. Cytocompatibility was assessed using HUVEC and human bMSC. Generally, the high compressability of the scaffold followed by restorage of its initial form allowed effective cell loading. Moreover, cells survived, kept their differentiation and remained proliferative as proved by various standard experiments.

The experiments were completed by two animal studies in order to analyse the value of the scaffold (with or without MSC, EC, MSC+EC) for treatment of large bone

defects, respectively for treatment of osteoporotic lesions thereby using rat models. Scaffold cylinders, either empty as control or loaded with cells were placed into the bone defect and vascularization and bone healing was assessed. It could be shown that vascularization was improved if EC were present, the vascular network reflects thereby impressively the architecture of the scaffold, demonstrating the beneficial effect of the transplanted EC on the formation of the new vasculature. Bone healing was assessed 12 weeks after operation. Constructs loaded with MSC, respectively MSC and EC lead to a good bone healing (increasing bone volume, trabecel density) whereas the cell free scaffold did not. Unfortunately biomechanical tests were not performed. Similar bone healing results were seen in the osteoporosis model.

All in all an extensive and comprehensive manuscript is provided. The data are original as proved by pubmed and google scholar research. No signs of data manipulation were obvious. The integration of small hollow tubes provides new, unique characteristics to the 'optically' conventional scaffold design which probably ease processes such as cell loading or fitting into bone defects. Despite these advantages, the regenerative potential seems in my eyes comparable to similarly designed 3D printed scaffolds.

Although a very well elaborated study is presented, some questions/suggestions remain as pointed out in the following.

Respond: We're deeply grateful for the reviewer's recognition and support of our work. The suggestion to conduct biomechanical tests to investigate bone healing is valuable. We regret that we were unable to include biomechanical testing in this study, but we will incorporate this constructive suggestion into our future studies for a more systematic investigation. According to the comments of the reviewer, we have carefully revised the manuscript. The point-by-point response to the reviewer's comments is as follows:

1. It is only a formal aspect but a page numbering would facilitate the review of the manuscript.

Respond: We have added a page number in the revised manuscript.

2. It can be assumed that the improved compressibility is a consequence of the hollow tubes. This should be more clearly stated in the manuscript.

Respond: We have added a clearer statement that the improved compressibility is a consequence of the hollow tubes in the revised manuscript (P7 L7-9).

3. Cell loading by repeated compression and suction: With the scaffold volume of 1 cubic centimeter, one compression cycle seems to be sufficient to draw enough cells also into the interior of the scaffold. Are multiple cycles necessary for larger volumes?

And: How quickly are the cells adherent, are they not flushed out of the material again by another compression cycle? Have you performed any analyses on this?

Respond: To illustrate the applicability of the post-bioprinting strategy for cell loading to larger sizes of HHSs, we have added the larger size of HHSs (15 × 15 × 10 mm and 20 × 10 × 10 mm) for cell loading. As shown in Supplementary Fig. 9, the larger size of HHSs (15 × 15 × 10 mm and 20 × 10 × 10 mm) could also rapidly load hBMSCs under a cycle of compression-recovery within 4 s at 80% strain. And, one compression cycle was sufficient to draw enough cells into the interior of the scaffold. In subsequent research, we will refine and identify the maximum size applicable for the cell loading method employed in this study.

Supplementary Fig. 9 Cell loading to larger size of HHSs (15 × 15 × 10 mm and 20 × 10 × 10 mm) at various regions with mechanical responsiveness.

As shown in Fig. 5d, the cell loading and compression cycles didn't exhibit a linear correlation, indicating that a part of cells would loss during continuously

compression cycles. The number of cell loading was 298 ± 64 cells/mm³ after one compression cycle, while 510 ± 114 cells/mm³ after three compression cycles. We think that if a cultivation period (such as 4 h) is introduced after the first compression cycle to allow cells to adhere to HHS, they are less likely to be flushed out during the second cycle. However, in this study, our goal was to achieve cell loading within a relatively short time, so we did not implement this step. Nonetheless, in future work, we plan to explore this approach more extensively.

4. Are cells also settled in the tube structures during cell loading by compression and subsequent suction of the cells? Based on the photographs (Fig. 5), this could be assumed. Is cell survival and proliferation altered within the tubes versus cells on the structures?

Respond: Cells can settle in the tube structures during cell loading by compression and subsequent suction of the cells without additional flushing of the grids with medium. Whether cell survival and proliferation altered within the tubes versus cells on the structures, is a highly thought-provoking question. As shown in Fig. 5a, and we have not observed significant difference presently, while whether there is slight difference will be deeply study in the future work.

5. Mechanical properties of HHSs: Are the numbers provided for “z” in the unit mm?

Respond: The numbers provided for z are in the unit mm, with “z” ranging from 0 to 0.6 mm, as described in P6 L8.

6. Cell viability under mechanical stimulation: What is the cell loss during compression cycles? Or does 40% compression did not lead to cell flushing out of the scaffold?

Respond: As shown in Fig. 5d, the cell loading and compression cycles didn't exhibit a linear correlation, indicating that a part of cells would loss during compression cycles. The number of cell loading was 298 ± 64 cells/mm³ after one compression

cycle, while 510 ± 114 cells/mm³ after three compression cycles.

7. P6, 116: avoid ratings such as “attractive” in the results section.

Respond: We have delete “attractive” in the results section in the revised manuscript.

8. P9, 123 (Materials and Methods section) source or strain of hBMSC-GFP?

Respond: hBMSCs-GFP was purchased from Cyagen, China (HUXMA-01101), which was described in P20 L21.

9. Discussion p15, 14-6: Scalability, is cell loading generally also applicable in larger scaffolds? How compression strain and cycles have to be adjusted?

Respond: To discuss the scalability of the post-bioprinting strategy for cell loading to larger sizes of HHSs, we firstly added the experimental results of larger size of HHSs for cell loading in the results part (P9 L12-14). “*The larger size of HHSs (15 ×15 ×10 mm and 20 ×10 ×10 mm) could also rapidly load hBMSCs under a cycle of compression-recovery within 4 s at 80% strain (Supplementary Fig. 9)*”. In discussion parts, we added the discussion “*Notably, the post-bioprinting strategy for cell loading can be extended to larger size of HHSs by increasing compression strain and cycles*”.

10. Figure 3b: Please remove the heart images. It is just an analogy and does not provide relevant new information.

Respond: We have removed the heart images in Fig. 3b in the revised manuscript.

11. Supplemental images: It is stated in results section (p5112) that “...formed tailed bridges between nanoclay structures”. Bridges are obviously seen but also in the control without nanoclay. To be honest I am not sure if I can see nanoclay structures. Please provide additional images in higher magnification or mark the structures that are supposed to be nanoclay structures.

Respond: The hybrid inks of gelatin methacryloyl/Laponite nanoclay/ N-acryloyl glycinamide, noted as GLNx (namely x = 0, 4, 8, and 12, respectively), N represents

N-acryloyl glycinamide rather than nanoclay, while the other components were kept constant (gelatin methacryloyl, 12% w/v; Laponite nanoclay, 10% w/v). Thus, all of groups contain Laponite nanoclay. To enhance the clarity of nanoclay, we have incorporated additional SEM images at a higher magnification in Supplementary Figure 3. The white arrows point to nanoclay.

12. Discussion: Please compare bone healing response with other, more ‘conventionally’ 3D printed scaffolds. In my opinion bone healing results are comparable to other solutions especially when considering the long period of 12 weeks.

Respond: We believe that it's necessary to compare the bone defect repair materials with existing research. However, to be frank, this task poses huge challenges due to variations in animals, models, sizes of bone defects, and standards across different studies. Thus, based on the same animal model (rat segmental bone defects), we have compared bone healing response of HHS-cells in the discussion parts (P16 L13-17) with others. “Whereas, in contrast to other cell-free constructs, such as 3D printed scaffolds, porous scaffolds and hydrogels, HHS-cells also exhibit a comparable repair effect in rats segmental bone defects^{3, 50-52}.”

In addition, we conducted a comparative analysis highlighting the advantages of post-bioprinting over other cell loading methods (Supplementary Fig. 19).

REVIEWERS' COMMENTS

Reviewer #1 (Remarks to the Author):

The authors have satisfactorily responded to my concerns with a thorough revision of the text and pointing to relevant data in main figures and SI. In my opinion, the manuscript is now acceptable for publication.

Reviewer #2 (Remarks to the Author):

The authors already answered all my previous concerns.

Reviewer #3 (Remarks to the Author):

A thoroughly revised manuscript was submitted. All of my issues and queries and, to my opinion, those of the other reviewers as well, were addressed satisfactorily.

As I stated in my initial review, this is a valuable research for the field of tissue engineering, which is now substantially improved.